# Fast and Provable ADMM for Learning with Generative Priors

**Fabian Latorre, Armin Eftekhari and Volkan Cevher**
Laboratory for information and inference systems (LIONS)
EPFL, Lausanne, Switzerland
{firstname.lastname}@epfl.ch

## Abstract

In this work, we propose a (linearized) Alternating Direction Method-of-Multipliers (ADMM) algorithm for minimizing a convex function subject to a nonconvex constraint. We focus on the special case where such constraint arises from the specification that a variable should lie in the range of a neural network. This is motivated by recent successful applications of Generative Adversarial Networks (GANs) in tasks like compressive sensing, denoising and robustness against adversarial examples. The derived rates for our algorithm are characterized in terms of certain geometric properties of the generator network, which we show hold for feedforward architectures, under mild assumptions. Unlike gradient descent (GD), it can efficiently handle non-smooth objectives as well as exploit efficient partial minimization procedures, thus being faster in many practical scenarios.

## 1 Introduction

Generative Adversarial Networks (GANs) [Goodfellow et al., 2014] show great promise for faithfully modeling complex data distributions, such as natural images [Radford et al., 2015, Brock et al., 2019] or audio signals [Engel et al., 2019, Donahue et al., 2019]. Understanding and improving the theoretical and practical aspects of their training has thus attracted significant interest [Lucic et al., 2018, Mescheder et al., 2018, Daskalakis et al., 2018, Hsieh et al., 2018, Gidel et al., 2019].

Researchers have also begun to leverage the modeling power of GANs and other generative models like Variational Auto-encoders [Kingma and Welling, 2013] in applications ranging from compressive sensing [Bora et al., 2017], to image denoising [Lipton and Tripathi, 2017, Tripathi et al., 2018], to robustness against adversarial examples [Ilyas et al., 2017, Samangouei et al., 2018].

These and other [Dhar et al., 2018, Ulyanov et al., 2018] applications model high-dimensional data as the output of the generator network associated with a generative model, and often lead to a highly non-convex optimization problem of the form $\min_z f(G(z))$, where the the generator $G$ is nonlinear and $f$ is convex. We then find the optimal *latent vector $z$*, as illustrated in Section 5 with several examples.

This optimization problem involving a generative model poses various difficulties for existing first-order algorithms. Indeed, to our knowledge, the only existing provable algorithm for solving (1) relies on the existence of a projection oracle, and is limited to the special case of *compressive sensing* with a generative prior [Shah and Hegde, 2018, Hegde, 2018], see Section 4 for the details. The main computational bottleneck is of course the non-convex projection step, for which no convergence analysis in terms of the geometry of the underlying generator $G$ currently exists.

On the other hand, Gradient Descent (GD) and its adaptive variants [Kingma and Ba, 2014] cannot efficiently handle non-smooth objective functions, as they are entirely oblivious to the composite structure of the problem [Nesterov, 2013b]. A simple example is denoising with the $\ell_\infty$-norm,

for which subgradient descent (as the standard non-smooth alternative to GD) fails in practice, as observed in Section 5.

With the explosion of generative models in popularity, there is consequently a pressing need for provable and flexible optimization algorithms to solve the resulting non-convex and (possibly) non-smooth problems. The present work addresses this need by focusing on the general optimization template

$$\begin{aligned} \underset{w,z}{\text{minimize}} \quad & F(w,z) := L(w) + R(w) + H(z) \\ \text{subject to} \quad & w = G(z), \end{aligned} \tag{1}$$

where $L : \mathbb{R}^d \to \mathbb{R}$ is convex and smooth, $R : \mathbb{R}^d \to \mathbb{R}$ and $H : \mathbb{R}^s \to \mathbb{R}$ are convex but not necessarily smooth, and $G : \mathbb{R}^s \to \mathbb{R}^d$ is differentiable but often non-linear, corresponding to the generator network associated with a generative model. Even though $R$ and $H$ might not be smooth, we assume throughout that their proximal mappings can be efficiently computed [Parikh et al., 2014].

For brevity, we refer to (1) as *optimization with a generative prior* whenever $G$ is given by the generator network associated with a generative model [Kingma and Welling, 2013, Goodfellow et al., 2014]. In this context, we make three key contributions, summarized below:

**1. Algorithm:** We propose an efficient and scalable (linearized) Alternating Direction Method-of-Multipliers (ADMM) framework to solve (1), see Algorithm 1. To our knowledge, this is the first non-convex and linearized ADMM algorithm for nonlinear constraints with provable fast rates to solve problem (1), see Section 4 for a detailed literature review.

We evaluate this algorithm numerically in the context of denoising with GANs in the presence of adversarial or stochastic noise, as well as compressive sensing [Bora et al., 2017]. In particular, Algorithm 1 allows for efficient denoising with the $\ell_\infty$- and $\ell_1$-norms, with applications in defenses against adversarial examples [Szegedy et al., 2013] and signal processing, respectively.

**2. Optimization guarantees:** We prove fast approximate convergence for Algorithm 1 under the assumptions of smoothness and near-isometry of $G$, as well as strong convexity of $L$. That is, we distill the key geometric attributes of the generative network $G$ responsible for the success of Algorithm 1. We then show how some common neural network architectures satisfy these geometric assumptions.

We also establish a close relation between a variant of Algorithm 1 and the gradient descent in [Bora et al., 2017] and, in this sense, provide the first rates for it, albeit in a limit case detailed in Section 3. Indeed, one key advantage of the primal-dual formulation studied in this paper is exactly this versatility, as well as the efficient handling of non-smooth objectives.

Lastly, we later relax the assumptions on $L$ to *restricted* strong convexity/smoothness, thus extending our results to the broader context of statistical learning with generative priors, which includes compressive sensing [Bora et al., 2017] as a special case.

**3. Statistical guarantees:** In the context of statistical learning with generative priors, where $L$ in (1) is replaced with an *empirical risk*, we provide the generalization error associated with Algorithm 1. That is, we use the standard notion of Rademacher complexity [Mohri et al., 2018] to quantify the number of training data points required for Algorithm 1 to learn the true underlying parameter $w^\natural$.

## 2 Algorithm

In this section, we adapt the powerful Alternating Descent Method of Multipliers (ADMM) [Glowinski and Marroco, 1975, Gabay and Mercier, 1976, Boyd et al., 2011] to solve the non-convex problem (1). We define the corresponding *augmented Lagrangian* with the dual variable $\lambda \in \mathbb{R}^p$ as

$$\mathcal{L}_\rho(w, z, \lambda) := L(w) + \langle w - G(z), \lambda \rangle + \frac{\rho}{2} \|w - G(z)\|_2^2, \tag{2}$$

for a penalty weight $\rho > 0$. By a standard duality argument, (1) is equivalent to

$$\min_{w,z} \max_\lambda \mathcal{L}_\rho(w, z, \lambda) + R(w) + H(z). \tag{3}$$

Applied to (3), every iteration of ADMM would minimize the augmented Lagrangian with respect to $z$, then with respect to $w$, and then update the dual variable $\lambda$. Note that $\mathcal{L}_\rho(w, z, \lambda)$ is often

non-convex with respect to $z$ due to the nonlinearity of the generator $G : \mathbb{R}^s \to \mathbb{R}^d$ and, consequently, the minimization step with respect to $z$ in ADMM is often intractable.

To overcome this limitation, we next *linearize* ADMM. In the following, we let $\mathrm{P}_R$ and $\mathrm{P}_H$ denote the *proximal maps* of $R$ and $H$, respectively [Parikh et al., 2014].

The equivalence of problems (1) and (3) motivates us to consider the following algorithm for the penalty weight $\rho > 0$, the primal step sizes $\alpha, \beta > 0$, and the positive dual step sizes $\{\sigma_t\}_{t \geq 0}$:

$$
\begin{aligned}
z_{t+1} &= \mathrm{P}_{\beta H} \left( z_t - \beta \nabla_z \mathcal{L}_\rho(w_t, z_t, \lambda_t) \right), \\
w_{t+1} &= \mathrm{P}_{\alpha R} \left( w_t - \alpha \nabla_w \mathcal{L}_\rho(w_t, z_{t+1}, \lambda_t) \right), \\
\lambda_{t+1} &= \lambda_t + \sigma_{t+1}(w_{t+1} - G(z_{t+1})).
\end{aligned}
\tag{4}
$$

As opposed to ADMM, to solve (1), the linearized ADMM in (4) takes only one descent step in both $z$ and $w$, see Algorithm 1 for the summary. The particular choice of the dual step sizes $\{\sigma_t\}_t$ in Algorithm 1 ensures that the dual variables $\{\lambda_t\}_t$ remain bounded, see [Bertsekas, 1976] for a precedent in the convex literature.

**Algorithm 2.** Let us introduce an important variant of Algorithm 1. In our setting, $\mathcal{L}_\rho(w, z, \lambda)$ is in fact convex with respect to $w$ and therefore Algorithm 2 replaces the first step in (4) with exact minimization over $w$. This exact minimization step can be executed with an off-the-shelf convex solver, or might sometimes have a closed-form solution. Moreover, Algorithm 2 gradually increases the penalty weight to emulate a multi-scale structure. More specifically, for an integer $K$, consider the sequences of penalty weights and primal step sizes $\{\rho_k, \alpha_k, \beta_k\}_{k=1}^{K}$, specified as

$$
\rho_k = 2^k \rho, \qquad \alpha_k = 2^{-k} \alpha, \qquad \beta_k = 2^{-k} \beta, \qquad k \leq K.
\tag{5}
$$

Consider also a sequence of integers $\{n_k\}_{k=1}^{K}$, where

$$
n_k = 2^k n, \qquad k \leq K,
\tag{6}
$$

for an integer $n$. At (outer) iteration $k$, Algorithm 2 executes $n_k$ iterations of Algorithm 1 with exact minimization over $w$. Then it passes the current iterates of $w$, $z$, and dual step size to the next (outer) iteration. Loosely speaking, Algorithm 2 has a multi-scale structure, allowing it to take larger steps initially and then slowing down as it approaches the solution. As discussed in Section 3, the theoretical guarantees for Algorithm 1 also apply to Algorithm 2. The pseudocode for Algorithm 2 is given in Supplementary I.

As the closing remark, akin to the convex case [He et al., 2000, Xu et al., 2017], it is also possible to devise a variant of Algorithm 1 with adaptive primal step sizes, which we leave for a future work.

---

**Algorithm 1** Linearized ADMM for solving problem (1)

**Input:** Differentiable $L$, proximal-friendly convex regularizers $R$ and $H$, differentiable prior $G$, penalty weight $\rho > 0$, primal step sizes $\alpha, \beta > 0$, initial dual step size $\sigma_0 > 0$, primal initialization $w_0$ and $z_0$, dual initialization $\lambda_0$, stopping threshold $\tau_c > 0$.

1 **for** $t = 0, 1, \ldots, T - 1$ **do**

2 $\qquad z_{t+1} \leftarrow \mathrm{P}_{\beta H} \left( z_t - \beta \nabla_z \mathcal{L}_\rho(w_t, z_t, \lambda_t) \right)$ $\hfill$ (primal updates)

3 $\qquad w_{t+1} \leftarrow \mathrm{P}_{\alpha R} \left( w_t - \alpha \nabla_w \mathcal{L}_\rho(w_t, z_{t+1}, \lambda_t) \right)$

4 $\qquad \sigma_{t+1} \leftarrow \min \left( \sigma_0, \dfrac{\sigma_0}{\|w_{t+1} - G(z_{t+1})\|_2 \, t \log^2(t+1)} \right)$ $\hfill$ (dual step size)

5 $\qquad \lambda_{t+1} \leftarrow \lambda_t + \sigma_{t+1}(w_{t+1} - G(z_{t+1}))$ $\hfill$ (dual update)

6 $\qquad s \leftarrow \dfrac{\|z_{t+1} - z_t\|_2^2}{\alpha} + \dfrac{\|w_{t+1} - w_t\|_2^2}{\beta} + \sigma_t \|w_t - G(z_t)\|_2^2 \leq \tau_c$ $\hfill$ (stopping criterion)

7 $\qquad$ **if** $s \leq \tau_c$ **then**

8 $\qquad\qquad$ **return** $(w_{t+1}, z_{t+1})$

9 **return** $(w_T, z_T)$

---

# 3 Optimization Guarantees

Let us study the theoretical guarantees of Algorithm 1 for solving program (1), whose constraints are nonlinear and non-convex (since $G$ is specified by a neural network). The main contribution of this section is Theorem 1, which is inherently an optimization result stating that Algorithm 1 succeeds under certain assumptions on (1).

From an optimization perspective, to our knowledge, Theorem 1 is the first to provide (fast) rates for non-convex and linearized ADMM, see Section 4 for a detailed literature review. The assumptions imposed below on $L$ and the generator $G$ ensure the success of Algorithm 1 and are shortly justified for our setup, where $G$ is a generator network.

**Assumption 1. strong convexity / smoothness of $L$:** *We assume that $L$ in (1) is both strongly convex and smooth, namely, there exist $0 < \mu_L \leq \nu_L$ such that*

$$\frac{\mu_L}{2}\|w - w'\|^2 \leq L(w') - L(w) - \langle w' - w, \nabla L(w) \rangle \leq \frac{\nu_L}{2}\|w - w'\|^2, \quad \forall w, w' \in \mathbb{R}^d. \quad (7)$$

Assumption 1 is necessary to establish fast rates for Algorithm 1, and is readily met for $L(w) = \|w - \widehat{w}\|_2^2$ with $\mu_L = \nu_L = 1$, which renders Algorithm 1 applicable to $\ell_2$-denoising with generative prior in [Tripathi et al., 2018, Samangouei et al., 2018, Ilyas et al., 2017]. Here, $\widehat{w}$ is the noisy image.

In Supplementary A, we also relax the strong convexity/smoothness in Assumption 1 to *restricted* strong convexity/smoothness, which enables us to apply Theorem 1 in the context of statistical learning with a generative prior, for example in compressive sensing [Bora et al., 2017].

Under Assumption 1, even though $L$ and consequently the objective function of (1) are strongly convex, problem (1) might *not* have a unique solution, which is in stark contrast with convex optimization. Indeed, a simple example is minimizing $x^2 + y^2$ with the constraint $x^2 + y^2 = 1$. We next state our assumptions on the generator $G$.

**Assumption 2. Strong smoothness of $G$:** *Let $DG$ be the Jacobian of $G$. We assume that $G : \mathbb{R}^s \to \mathbb{R}^d$ is strongly smooth, namely, there exists $\nu_G \geq 0$ such that*

$$\|G(z') - G(z) - DG(z) \cdot (z' - z)\|_2 \leq \frac{\nu_G}{2}\|z' - z\|_2^2, \qquad \forall z, z' \in \mathbb{R}^s, \quad (8)$$

**Assumption 3. Near-isometry of $G$:** *We assume that the generative prior $G$ is a near-isometric map, namely, there exist $0 < \iota_G \leq \kappa_G$ such that*

$$\iota_G\|z' - z\|_2 \leq \|G(z') - G(z)\|_2 \leq \kappa_G\|z' - z\|_2, \qquad \forall z, z' \in \mathbb{R}^s. \quad (9)$$

The invertibility of certain network architectures have been established before in [Ma et al., 2018, Hand and Voroninski, 2017]. More concretely, Assumptions 2 and 3 hold for a broad class of generators, as summarized in Proposition 1 and proved in Supplementary B.

**Proposition 1.** *Let $G_\Xi : \mathcal{D} \subset \mathbb{R}^d \to \mathbb{R}^s$ be a feedforward neural network with weights $\Xi \in \mathbb{R}^h$, $k$ layers, non-decreasing layer sizes $s \leq s_1 \leq \ldots s_k \leq d$, with $\omega_i$ as activation function in the $i$-th layer, and compact domain $\mathcal{D}$. For every layer $i$, suppose that the activation $\omega_i : \mathbb{R} \to \mathbb{R}$ is of class $C^1$ (continuously-differentiable) and strictly increasing. Then, after an arbitrarily small perturbation to the weights $\Xi$, Assumptions 2 and 3 hold almost surely with respect to the Lebesgue measure.*

A few comments about the preceding result are in order.

**Choice of the activation function:** Strictly-increasing $C^1$ activation functions in Proposition 1, such as the Exponential Linear Unit (ELU) [Clevert et al., 2015] or softplus [Dugas et al., 2001], achieve similar or better performance compared to the commonly-used (but non-smooth) Rectified Linear Activation Unit (ReLU) [Xu et al., 2015, Clevert et al., 2015, Gulrajani et al., 2017, Kumar et al., 2017, Kim et al., 2018].

In our experiments in Section 5, we found that using ELU activations for the generator $G$ does not adversely affect the representation power of the trained generator. Lastly, the activation function for the final layer of the generator is typically chosen as the sigmoid or tanh [Radford et al., 2015], for which the conditions in Proposition 1 are also met.

**Compact domain:** The compactness requirement in Proposition 1 is mild. Indeed, even though the Gaussian distribution is the default choice as the input for the generator in GANs, training has also

been successful using compactly-supported distributions, such as the uniform distribution [Lipton and Tripathi, 2017].

Interestingly, even after training with Gaussian noise, limiting the resulting generator to a truncated Gaussian distribution can in fact boost the performance of GANs [Brock et al., 2019], as measured with common metrics like the Inception Score [Salimans et al., 2016] or Frechet Inception Distance [Heusel et al., 2017]. This evidence suggests that obtaining a good generator $G$ with compact domain is straightforward. In the experiments of Section 5, we use truncated Gaussian on an Euclidean ball centered at the origin.

**Non-decreasing layer sizes:** This is a standard feature of popular generator architectures such as the DCGAN [Radford et al., 2015] or infoGAN [Chen et al., 2016]. This property is also exploited in the analysis of the optimization landscape of problem (1) by Hand and Voroninski [2017], Heckel et al. [2019] and for showing invertiblity of (de)convolutional generators [Ma et al., 2018].

**Necessity of assumptions on $G$:** Assumptions 2 and 3 on the generator $G$ are necessary for the provable success of Algorithm 1. Loosely speaking, Assumption 2 controls the curvature of the generative prior, without which the dual iterations can oscillate without improving the objective.

On the other hand, the lower bound in (9) means that the generative prior $G$ must be *stably* injective: Faraway latent parameters should be mapped to faraway outputs under $G$. As a pathological example, consider the parametrization of a circle as $\{(\sin z, \cos z) : z \in [0, 2\pi)\}$.

This stable injectivity property in (9) is necessary for the success of Algorithm 1 and is not an artifact of our proof techniques. Indeed, without this condition, the $z$ updates in Algorithm 1 might not reduce the feasibility gap $\|w - G(z)\|_2$. Geometric assumptions on nonlinear constraints have precedent in the optimization literature [Birgin et al., 2016, Flores-Bazán et al., 2012, Cartis et al., 2018] and to a lesser extent in the literature of neural networks too [Hand and Voroninski, 2017, Ma et al., 2018], which we further discuss in Section 4.

Having stated and justified our assumptions on $L$ and the generator $G$ in (1), we are now prepared to present the main technical result of this section. Theorem 1 states that Algorithm 1 converges linearly to a small neighborhood of a solution, see Supplementary C for the proof.

**Theorem 1. (guarantees for Algorithm 1)** *Suppose that Assumptions 1-3 hold. Let $(w^*, z^*)$ be a solution of program (1) and let $\lambda^*$ be a corresponding optimal dual variable. Let also $\{w_t, z_t, \lambda_t\}_{t \geq 0}$ denote the output sequence of Algorithm 1. Suppose that the primal step sizes $\alpha, \beta$ satisfy*

$$\alpha \leq \frac{1}{\nu_\rho}, \qquad \beta \leq \frac{1}{\xi_\rho + 2\alpha\tau_\rho^2}. \qquad \sigma_0 \leq \sigma_{0,\rho}. \tag{10}$$

*Then it holds that*

$$\frac{\|w_t - w^*\|_2^2}{\alpha} + \frac{\|z_t - z^*\|_2^2}{\beta} \leq 2(1 - \eta_\rho)^t \Delta_0 + \frac{\overline{\eta}_\rho}{\rho}, \tag{11}$$

$$\|w_t - G(z_t)\|_2^2 \leq \frac{4(1 - \eta_\rho)^t \Delta_0}{\rho} + \frac{\widetilde{\eta}_\rho}{\rho^2}, \tag{12}$$

*for every iteration t. Above, $\Delta_0 = \mathcal{L}_\rho(w_0, z_0, \lambda_0) - \mathcal{L}_\rho(w^*, z^*, \lambda^*)$ is the initialization error, see (2). The convergence rate $1 - \eta_\rho \in (0, 1)$ and the quantities $\nu_\rho, \xi_\rho, \tau_\rho, \sigma_{0,\rho}, \overline{\eta}_\rho, \widetilde{\eta}_\rho$ above depend on the parameters in Assumptions 1-3 and on $\lambda^*$, as specified in the proof. As an example, in the regime where $\mu_L \gg \rho$ and $\iota_G^2 \gg \nu_G$, we can take*

$$\alpha \approx \frac{1}{\nu_L}, \qquad \beta \approx \frac{1}{\rho\kappa_G^2}, \qquad \frac{\rho\nu_G}{\kappa_G^2} \lesssim \sigma_0 \lesssim \rho \min\left(\frac{\mu_L^2}{\nu_L^2}, \frac{\iota_G^4}{\kappa_G^4}\right),$$

$$\eta_\rho \approx \min\left(\frac{\mu_L}{\nu_L}, \frac{\iota_G^2}{\kappa_G^2}\right), \qquad \overline{\eta}_\rho \approx \widetilde{\eta}_\rho \approx \max\left(\frac{\nu_L}{\mu_L}, \frac{\kappa_G^2}{\iota_G^2}\right). \tag{13}$$

*Above, for the sake of clarity, $\approx$ and $\lesssim$ suppress the universal constants, dependence on the initial dual $\lambda_0$ and the corresponding step size $\sigma_0$.*

A few clarifying comments about Theorem 1 are in order.

**Error:** According to Theorem 1, if the primal and dual step sizes are sufficiently small and Assumptions 1-3 are met, Algorithm 1 converges linearly to a *neighborhood* of a solution $(w^*, z^*)$. The size of this neighborhood depends on the penalty weight $\rho$ in (2). For instance, in the example in Theorem 1, it is easy to verify that this neighborhood has a radius of $O(1/\rho)$, which can be made smaller by increasing $\rho$.

Theorem 1 is however silent about the behavior of Algorithm 1 within this neighborhood. This is to be expected. Indeed, even in the simpler convex case, where $G$ in program (1) would have been an affine map, provably no first-order algorithm could converge linearly to the solution [Ouyang and Xu, 2018, Agarwal et al., 2010].

Investigating the behavior of Algorithm 1 within this neighborhood, while interesting, arguably has little practical value. For example, in the convex case, ADMM would converge slowly (sublinearly) in this neighborhood, which does not appeal to the practitioners.

As another example, when Algorithm 1 is applied in the context of statistical learning, there is no benefit in solving (1) beyond the statistical accuracy of the problem at hand [Agarwal et al., 2010], see the discussion in Supplementary A.1. As such, we defer the study of the local behavior of Algorithm 1 to a future work.

**Feasibility gap:** Likewise, according to (24) in Theorem 1, the feasibility gap of Algorithm 1 rapidly reaches a plateau. In the example in Theorem 1, the feasibility gap rapidly reaches $O(1/\rho)$, where $\rho$ is the penalty weight in (2). As before, even in the convex case, no first-order algorithm could achieve exact feasibility at linear rate [Ouyang and Xu, 2018, Agarwal et al., 2010].

**Intution:** While the exact expressions for the quantities in Theorem 1 are given in Supplementary C, the example provided in Theorem 1 highlights the simple but instructive regime where $\mu_L \gg \rho$ and $\iota_G^2 \gg \nu_G$, see Assumptions 1-3. Intuitively, $\mu_L \gg \rho$ means that minimizing the objective of (1) is prioritized over reducing the feasibility gap, see (2). In addition, $\iota_G^2 \gg \nu_G$ suggests that the generative prior $G$ is very smooth.

In this regime, the primal step size $\alpha$ for $w$ updates is determined by how smooth $L$ is, and the primal step size $\beta$ in the latent variable $z$ is determined by how smooth $G$ is, see (13). Similar restrictions are standard in first-order algorithms to avoid oscillations [Nesterov, 2013a].

As discussed earlier, the algorithm rapidly reaches a neighborhood of size $O(1/\rho)$ of a solution and the feasibility gap plateaus at $O(1/\rho)$. Note the trade-off here for the choice of $\rho$: the larger the penalty weight $\rho$ is, the more accurate Algorithm 1 would be and yet increasing $\rho$ is restricted by the assumption $\rho \ll \mu_L$. Moreover, in this example, the rate $1 - \eta_\rho$ of Algorithm 1 depends only on the regularity of $L$ and $G$ in program (1), see (13). Indeed, the more well-conditioned $L$ is and the more near-isometric $G$ is, the larger $\eta_\rho$ and the faster the convergence would be.

Generally speaking, increasing the penalty weight $\rho$ reduces the bias of Algorithm 1 at the cost of a slower rate. Beyond our work, such dependence on the geometry of the constraints has precedent in the literature of optimization [Birgin et al., 2016, Flores-Bazán et al., 2012, Cartis et al., 2018] and manifold embedding theory [Eftekhari and Wakin, 2015, 2017].

**Relation to simple gradient descent:**   Consider a variant of Algorithm 1 that replaces the linearized update for $w$ in (4) with exact minimization with respect to $w$, which can be achieved with an off-the-shelf convex solver or might have a closed-form solution in some cases. The exact minimization over $w$ and Lemma 7 together guarantee that Theorem 1 also applies to this variant of Algorithm 1.

Moreover, as a special case of (1) where $R \equiv 0$ and $H \equiv 0$, this variant is closely related to GD [Bora et al., 2017], presented there without any rates. In Appendix F, we establish that the updates of both algorithms match as the feasibility gap vanishes.

In this sense, Theorem 1 provides the first rates for GD, albeit in the limit case of vanishing feasibility gap. Indeed, one key advantage of the primal-dual formulation studied in this paper is exactly this versatility in providing a family of algorithms, such as Algorithms 1 and 2, that can be tuned for various scenarios and can also efficiently handle the non-smooth case where $R$ or $H$ are nonzero in (1).

# 4 Related Work

Bora et al. [2017] empirically tune gradient descent for compressive sensing with a generative prior

$$\min_z \|A \cdot G(z) - b\|_2^2, \tag{14}$$

which is a particular case of template (1) (without splitting). They also provide a statistical generalization error dependent on a certain *set restricted isometry property* on the matrix $A$. More generally, Theorem 4 in Supplementary A provides statistical guarantees for Algorithm 1 using the standard notion of empirical Rademacher complexity [Mohri et al., 2018].

Hand and Voroninski [2017] analyze the optimization landscape of (14) under the assumption that $G$ $(i)$ is composed of linear layers and ReLU activation functions, $(ii)$ is sufficiently expansive at each layer and $(iii)$ the network's weights have a Gaussian distribution or an equivalent deterministic *weight distribution condition*. Under such conditions, they show global existence of descent directions outside small neighborhoods around two points, but do not provide algorithmic convergence rates. Their analysis requires ReLU activation in all layers of the generator $G$, including the last one, which is often not met in practice.

On the other hand, our framework is not restricted to a particular network architecture and instead isolates the necessary assumptions on the network $G$ for the success of Algorithm 1. In doing so, we effectively decouple the learning task from the network structure $G$ and study them separately in Theorem 1 and Proposition 1, respectively. In particular, our theory in Section 3 (Supplementary A) applies broadly to any nonlinear map $G$ that meets Assumptions 1-3 (Assumptions 2-5), respectively.

In turn, Proposition 1 establishes that the standard feed forward network with common differentiable activation functions almost surely meets these assumptions. In this sense, let us also point to the work of Oymak et al. [2018], which is limited to linear regression with a nonlinear constraint, with its convex analogue studied in [Agarwal et al., 2010, Giryes et al., 2016].

Heckel et al. [2019] provides a convergence proof for a modified version of gradient descent, limited to (14) and without specifying a rate. We provide the convergence rate for a broad range of learning problems, and study the statistical generalization. Hand et al. [2018] studied the *phase retrieval* problem, with a non-convex objective function that is not directly covered by (1).

For the problem (14), Shah and Hegde [2018], Hegde [2018] proposed to use Projected Gradient Descent (PGD) after splitting in a manner similar to our template (1). If the projection (onto the range of the prior $G$) is successful, and under certain additional conditions, the authors establish linear convergence of PGD to a minimizer of (14). However, the projection onto the nonlinear range of $G$ is itself a difficult non-convex program without any theoretical guarantees. In contrast, we can solve the same problem without any projections while still providing a convergence rate.

From an optimization perspective, there are no fast rates for linearized ADMM with nonlinear constraints to our knowledge, but convergence to a first-order stationary point and special cases in a few different settings have been studied [Liu et al., 2017, Shen et al., 2016, Chen and Gu, 2014, Qiao et al., 2016]. Let us again emphasize that Assumptions 2 and 3 extract the key attributes of $G$ necessary for the success of Algorithm 1, which is therefore not limited to a generator network. It is also worth noting another line of work that applies tools from statistical physics to inference with deep neural networks, see [Manoel et al., 2017, Rezende et al., 2014] and the references therein.

# 5 Experiments

In this section we evaluate our algorithms for image recovery tasks with a generative prior. The datasets we consider are the CelebA dataset of face images [Liu et al., 2015] and the MNIST dataset of handwritten digits [LeCun and Cortes, 2010]. We train a generator $G$ with ELU activation functions Clevert et al. [2015], in order to satisfy Assumption 2. The generators are trained using the Wasserstein GAN framework [Arjovsky et al., 2017]. For the CelebA dataset we downsample the images to $64 \times 64$ pixels as in Gulrajani et al. [2017] and we use the same residual architecture [He et al., 2015] for the generator with four residual blocks followed by a convolutional layer. For MNIST, we use the same architecture as one in Gulrajani et al. [2017], which contains one fully connected layer followed by three deconvolutional layers.

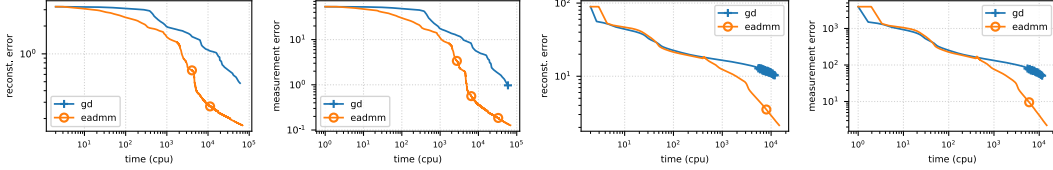

Figure 1: Reconstruction error and measurement error vs time (one tick equals the time of one GD iteration). MNIST (left) and CelebA (right).

We recover images on the range of the generator $G$, by choosing $z^\star \in \mathbb{R}^s$ and setting $w^\star := G(z^\star)$ as the true image to be recovered. This sets the global minimum of our objective functions at zero, and allows us to illustrate and compare the convergence rates of various algorithms.

Our Algorithm 1 mantains iterates $\{w_t, z_t\}_t$ where $w_t$ might not be feasible, namely, $w_t$ might not be in the range of $G$. As the goal in the following tasks is to recover an element in the range of $G$ (feasible points of (1)), we plot the objective value at the point $G(z_t)$.

**Baseline.** We compare to the most widely-used algorithm in the current literature, the gradient descent algorithm (GD) as used in [Bora et al., 2017], where a fixed number of iterations with constant step size are performed for the function $L(G(z))$. We tune its learning rate to be as large as possible without *overshooting*. (See Supplementary H for details on the hyperparameter tuning).

Our goal is to illustrate our theoretical results and highlight scenarios where Algorithm 1 can have better performance than GD in optimization problems with a generative prior. Hence, we do not compare with sparsity-prior based algorithms, such as LASSO [Tibshirani, 1996], or argue about GAN vs. sparsity priors as in Bora et al. [2017].

**Our algorithms.** We will use $(i)$ (linearized) ADMM (Algorithm 1), and $(ii)$ ADMM with exact minimization (Algorithm 2 a.k.a. EADMM), described in Section 2. For both ADMM and EADMM, we choose a starting iterate (random $z_0$ and $w_0 = G(z_0)$) and initial dual variable $\lambda_0 = 0$ (for GD we choose the same $z_0$ as initial iterate). We carefully track the objective function value vs. computation time for a fair comparison.

**Compressive sensing** The exact minimization step of EADMM involves the solution of a system of linear equations in each iteration. Performing Singular Value Decomposition (SVD) once on the measurement matrix $A$, and storing its components in memory, allows us to solve such linear systems with a very low per-iteration complexity (see Supplementary H.3). We plot the objective function value as well as the reconstruction error with $50\%$ relative measurements in Figure 1(average over 20 images (MNIST) and 10 images (CelebA)).

**Adversarial Denoising with $\ell_\infty$-norm** Projection onto the range of a deep-net prior has been considered by Samangouei et al. [2018], Ilyas et al. [2017] as a defense mechanism against adversarial examples [Szegedy et al., 2013]. In their settings, samples are denoised with a generative prior, before being fed to a classifier. Even though the adversarial noise introduced is typically bounded in $\ell_\infty$-norm, the projection is done in $\ell_2$-norm. Such projection corresponds to $F(w, z) = \|w - w^\natural\|^2$ in (1).

We instead propose to project using the $\ell_\infty$-norm that bounds the adversarial perturbation. To this end we let $F(w, z) = \gamma\|w - w^\natural\|_2^2 + \|w - w^\natural\|_\infty$ in the template (1), for some small value of $\gamma$. The proximal of the $\ell_\infty$ norm is efficiently computable [Duchi et al., 2008], allowing us to split $F(w, z)$ in its components $L(w) = \gamma\|w - w^\natural\|_2^2$ and $R(w) = \|w - w^\natural\|_\infty$ (Note that the small $\gamma$ ensures that Assumption 1 holds)

We compare the ADAM optimizer [Kingma and Ba, 2014], GD and ADMM (450 iterations and for GD and ADAM, and 300 iterations for EADMM). We use ADAM to solve the $\ell_2$ projection, while ADMM solves the $\ell_\infty$ projection. We evaluate on a test set of 2000 adversarial examples from the MNIST dataset, obtained with the Projected Gradient Algorithm of Madry et al. [2018] with 30 iterations, stepsize 0.01 and attack size 0.2. For the classifier, we use a standard convolutional network trained on clean MNIST samples. We also test ADAM, GD (3000 iterations) and EADMM (2000 iterations) on the $\ell_\infty$ denoising task.

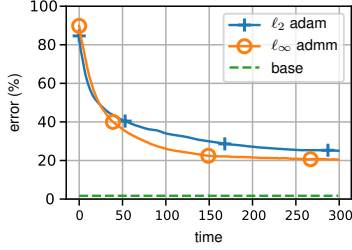

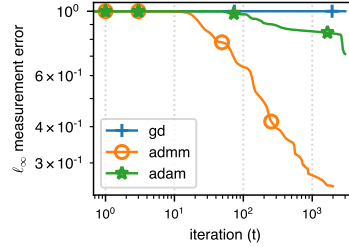

Figure 2: Test error on denoised adversarial examples vs computation time (average cpu time(s) over the sample).

Figure 3: $\ell_\infty$ reconstruction error per iteration for ADAM, GD, and EADMM.

The test error as a function of computation time is in Figure 2. We observe that the $\ell_\infty$ denoising performs better when faced with $\ell_\infty$ bounded attacks, in the sense that it achieves a lower error with less computation time. In Figure 3, we plot the $\ell_\infty$ reconstruction error achieved by ADAM, GD and EADMM, averaged over 7 images. GD was unable to decrease the initial error, while ADAM takes a considerable number of iterations to do so. In contrast, our ADMM already achieves the final error of ADAM within its first 100 iterations.

## 6 Conclusions and Future Work

In this work, we have proposed a flexible linearized ADMM algorithm for the minimization of a convex function subject to a nonlinear constraint given by a neural network. Under mild assumptions we demonstrate a fast convergence rate to a neighborhood of a solution of its Lagrangian formulation (3) (Theorem 1). Empirical evaluation shows how it can handle non-smooth terms more efficiently when compared to gradient descent and its variants.

Some avenues of research are left open which could yield faster variants of our proposed approach. First, ADMM-type algorithms admit acceleration and restart schemes with faster convergence rates in the convex case [Goldstein et al., 2014] but their adaptation to the nonlinear constraint given by a neural network is non-trivial. Secondly, adaptivity in the choice of penalty parameter $\rho$ can potentially improve the performance of the method and reduce the need for tuning [He et al., 2000]. Finally, the denoising with $\ell_\infty$-norm shows promise as a defense against adversarial examples, and its performance on higher dimensional datasets is worth investigating.

## Acknowledgements

This project has received funding from the European Research Council (ERC) under the European Union's Horizon 2020 research and innovation programme (grant agreement 725594 - time-data), the Department of the Navy - Office of Naval Research (ONR) under a grant number N62909-17-1-2111, and from the Swiss National Science Foundation (SNSF) under grant number 200021_178865. FL is supported through a PhD fellowship of the Swiss Data Science Center, a joint venture between EPFL and ETH Zurich. VC acknowledges the 2019 Google Faculty Research Award.

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
