[Supplementary Material]

# A Statistical Learning with Generative Priors

So far, we have assumed $L$ to be strongly convex in (1), see Assumption 1 and Theorem 1. In this section, we relax this assumption on $L$ in the context of statistical learning with generative priors, thus extending Theorem 1 to applications such as compressive sensing. We also provide the corresponding generalization error in this section.

Here, we follow the standard setup in learning theory Mohri et al. [2018]. Consider the probability space $(\mathbb{X}, \chi)$, where $\mathbb{X} \subset \mathbb{R}^d$ is a compact set, equipped with the Borel sigma algebra, and $\chi$ is the corresponding probability measure. To learn an unknown parameter $w^\natural \in \mathbb{R}^d$, consider the optimization program

$$\min_{w \in \mathbb{R}^p} L(w), \qquad L(w) := \mathbb{E}_{x \sim \chi} l(w, x), \tag{15}$$

where $L : \mathbb{R}^p \to \mathbb{R}$ is the differentiable *population risk* and $l : \mathbb{R}^d \times \mathbb{R}^p \to \mathbb{R}$ is the corresponding *loss function*. We also assume that Program (15) has a unique solution $w^\natural \in \mathbb{R}^p$. The probability measure $\chi$ above is itself often unknown and we instead have access to $m$ samples drawn independently from $\chi$, namely, $\{x_i\}_{i=1}^m \sim \chi$. This allows us to form the *empirical loss*

$$L_m(w) := \frac{1}{m} \sum_{i=1}^m l(w, x_i). \tag{16}$$

Often, $m \ll p$ and to avoid an ill-posed problem, we must leverage any inherent structure in $w^\natural$. In this work, we consider a differentiable map $G : \mathbb{R}^s \to \mathbb{R}^d$ and we assume that $w^\natural \in G(\mathbb{R}^s)$. That is, there exists $z^\natural \in \mathbb{R}^s$ such that $w^\natural = G(z^\natural)$. While not necessary, we limit ourselves in this section to the important case where $G$ corresponds to a neural network, see Section 1.

To learn $w^\natural$ with the generative prior $w^\natural = G(z^\natural)$, we propose to solve the program

$$\begin{aligned} \underset{w,z}{\text{minimize}} \qquad & L_m(w) + R(w) + H(z) \\ \text{subject to} \qquad & w = G(z), \end{aligned} \tag{17}$$

where $R : \mathbb{R}^p \to \mathbb{R}$ and $H : \mathbb{R}^s \to \mathbb{R}$ are convex but not necessarily smooth. Depending on the specific problem at hand, the *regularizers* $R$ and $H$ allow us to impose additional structure on $w$ and $z$, such as sparsity or set inclusion. Throughout, we again require that the proximal maps [Parikh et al., 2014] for $R$ and $H$ can be computed efficiently, as detailed in Section 2.

Let us now state our assumptions, some of which differ from Section 3.

**Assumption 4. Convexity / strong smoothness of loss:** *We assume that $l(\cdot, \cdot)$ is convex in both of its arguments. Moreover, we assume that $l(w, \cdot)$ is strongly smooth, namely, there exists $\sigma_l \geq 0$ such that for every $x, x' \in \mathbb{X}$*

$$D_l(x, x'; w) \leq \frac{\sigma_l}{2} \|x - x'\|_2^2, \tag{18}$$

*where $D_l$ stands for the* Bregman divergence *associated with $l(w, \cdot)$,*

$$D_l(x, x'; w) = l(w, x') - l(w, x) - \langle x' - x, \nabla_x l(w, x) \rangle.$$

**Assumption 5. Strong convexity / smoothness of the population risk:** *We assume that the population risk $L$ defined as*

$$L(w) := \mathbb{E}_{x \sim \chi} l(w, x), \tag{19}$$

*is both strongly convex and smooth, i.e., there exist $0 < \zeta_L \leq \sigma_L$ such that*

$$\frac{\zeta_L}{2} \|w - w'\|^2 \leq D_L(w, w') \leq \frac{\sigma_L}{2} \|w - w'\|^2,$$

$$D_L(w, w') = L(w') - L(w) - \langle w' - w, \nabla L(w) \rangle, \tag{20}$$

*for every $w, w' \in \mathbb{R}^d$. In the following we denote by $w^\natural$ the minimizer of (19). In view of our assumption, such minimizer is unique.*

Assumptions 4 and 5 are standard in statistical learning Mohri et al. [2018]. For example, in linear regression, we might take

$$l(w, x) = \frac{1}{2} |\langle w - w^\natural, x \rangle|^2,$$

$$L_m(w) = \frac{1}{2m} \sum_{i=1}^{m} |\langle w - w^\natural, x_i \rangle|^2,$$

for which both Assumptions 4 and 5 are met. Lastly, we require that the Assumptions 2 and 3 on $G$ hold in this section, see and Proposition 1 for when these assumptions hold for generative priors.

As a consequence of Assumption 4, we have that $L_m$ is convex. We additionally require $L_m$ to be strongly convex and smooth in the following restricted sense. Even though $L_m$ is random because of its dependence on the random training data $\{x_i\}_{i=1}^m$, we ensure later in this section that the next condition is indeed met with high probability when $m$ is large enough.

**Definition 1. Restricted strong convexity / smoothness of empirical loss:** *We say that $L_m$ is strongly convex and smooth on the set $W \subset \mathbb{R}^p$ if there exist $0 < \mu_L \le \nu_L$ and $\overline{\mu}_L, \overline{\nu}_L \ge 0$ such that*

$$D_{L_m}(w, w') \ge \frac{\mu_L}{2} \|w' - w\|_2^2 - \overline{\mu}_L,$$

$$D_{L_m}(w, w') \le \frac{\nu_L}{2} \|w' - w\|_2^2 + \overline{\nu}_L, \tag{21}$$

$$D_{L_m}(w, w') := L_m(w') - L_m(w) - \langle w' - w, \nabla L_m(w) \rangle,$$

*for every $w, w' \in W$.*

Under the above assumptions, a result similar to Theorem 1 holds, which we state without proof.

**Theorem 2. (guarantees for Algorithm 1)** *Suppose that Assumptions 2-5 hold. Let $(w^*, z^*)$ be a solution of program (1) and let $\lambda^*$ be a corresponding optimal dual variable. Let also $\{w_t, z_t, \lambda_t\}_{t \ge 0}$ denote the output sequence of Algorithm 1. Suppose that $L_m$ satisfies the restricted strong convexity and smoothness in Definition 1 for a set $W \subset \mathbb{R}^p$ that contains a solution $w^*$ of (1) and all the iterates $\{w_t\}_{t \ge 0}$ of Algorithm 1.[1] Suppose also that the primal step sizes $\alpha, \beta$ in Algorithm 1 satisfy*

$$\alpha \le \frac{1}{\nu_\rho}, \qquad \beta \le \frac{1}{\xi_\rho + 2\alpha\tau_\rho^2}. \qquad \sigma_0 \le \sigma_{0,\rho}, \tag{22}$$

*Then it holds that*

$$\frac{\|w_t - w^*\|_2^2}{\alpha} + \frac{\|z_t - z^*\|_2^2}{\beta} \le 2(1 - \eta_\rho)^t \Delta_0 + \frac{\overline{\eta}_\rho}{\rho}, \tag{23}$$

$$\|w_t - G(z_t)\|_2^2 \le \frac{4(1 - \eta_\rho)^t \Delta_0}{\rho} + \frac{\widetilde{\eta}_\rho}{\rho^2}, \tag{24}$$

*for every iteration $t$. Above, $\Delta_0 = \mathcal{L}_\rho(w_0, z_0, \lambda_0) - \mathcal{L}_\rho(w^*, z^*, \lambda^*)$ is the initialization error, see (2). The convergence rate $1 - \eta_\rho \in (0, 1)$ and the quantities $\nu_\rho, \xi_\rho, \tau_\rho, \sigma_{0,\rho}, \overline{\eta}_\rho, \widetilde{\eta}_\rho$ above depend on the parameters in the Assumptions 2-5 and on $\lambda_0, \sigma_0$.*

The remarks after Theorem 1 apply here too.

## A.1 Generalization Error

Building upon the optimization guarantee in Theorem 4, our next result in this section is Theorem 4, which quantifies the convergence of the iterates $\{w_t\}_{t \ge 0}$ of Algorithm 1 to the true parameter $w^\natural$.

In other words, Theorem 4 below controls the generalization error of (1), namely, the error incurred by using the empirical risk $L_m$ in lieu of the population risk $L$. Indeed, Theorem 1 is silent about $\|w_t - w^\natural\|_2$. We address this shortcoming with the following result, proved in Section G of the supplementary material.

**Lemma 3.** *Let $R = 1_W$ be the indicator function on $W \subset \mathbb{R}^p$ and set $H = 0$ in (1).[2] Suppose that $w^*$ belongs to the relative interior of $W$. Then it holds that*

$$\|w^\natural - w^*\|_2 \leq \frac{1}{\zeta_L} \max_{w \in W} \|\nabla L_m(w) - \nabla L(w)\|_2. \tag{25}$$

Before bounding the right-hand side of (25), we remark that it is possible to extend Lemma 3 to the case where the regularizer $R$ is a *decomposable* norm, along the lines of Negahban et al. [2012]. We will however not pursue this direction in the present work. Next note that (23) and Lemma 3 together imply that

$$
\begin{aligned}
\frac{\|w_t - w^\natural\|_2^2}{\alpha^2} &\leq \left( \frac{\|w_t - w^*\|_2}{\alpha} + \frac{\|w^* - w^\natural\|_2}{\beta} \right)^2 \quad &\text{(triangle inequality)} \\
&\leq \frac{2\|w_t - w^*\|_2^2}{\alpha^2} + \frac{2\|w^* - w^\natural\|_2^2}{\beta^2} \quad &((a+b)^2 \leq 2a^2 + 2b^2) \\
&\leq 4(1 - \eta_\rho)^t \Delta_0 + \frac{2\overline{\eta}_\rho}{\rho} + \frac{2}{\zeta_L^2} \max_{w \in W} \|\nabla L_m(w) - \nabla L(w)\|_2^2. &\tag{26}
\end{aligned}
$$

According to Theorem 1, the right-hand side of (26) depends on $\mu_L, \overline{\mu}_L, \nu_L, \overline{\nu}_L$, which were introduced in Definition 1. Note that $\mu_L, \overline{\mu}_L, \nu_L, \overline{\nu}_L$ and the right-hand side of (25) are all random variables because they depend on $L_m$ and thus on the randomly drawn training data $\{x_i\}_{i=1}^m$. To address this issue, we apply a basic result in statistical learning theory as follows. For every $w \in \mathbb{R}^p$ and every pair $x, x' \in \mathbb{X}$, we use Assumption 4 to write that

$$
\begin{aligned}
\|\nabla l(w, x) - \nabla l(w, x')\|_2 &\leq \sigma_l \|x - x'\|_2 \quad &\text{(see (18))} \\
&\leq \sigma_l \text{diam}(\mathbb{X}), &\tag{27}
\end{aligned}
$$

where $\text{diam}(\mathbb{X})$ denotes the diameter of the compact set $\mathbb{X}$. Note also that

$$\mathbb{E}_{\{x_i\}_i} \nabla L_m(w) = \nabla L(w), \qquad \forall w \in W, \tag{28}$$

where the expectation is over the training data $\{x_i\}_i$. Then, for $\varepsilon > 0$ and except with a probability of at most $e^{-\varepsilon}$, it holds that

$$
\begin{aligned}
\|\nabla L_m(w) &- \nabla L(w)\|_2 \\
&\leq 2\mathcal{R}_W(x_1, \cdots, x_m) + 3\sigma_l \text{diam}(\mathbb{X}) \sqrt{\frac{\varepsilon + 2}{2m}} \\
&=: \Upsilon_{m,W}(\varepsilon), &\tag{29}
\end{aligned}
$$

for every $w \in W$ [Mohri et al., 2018]. Above,

$$\mathcal{R}_W(x_1, \cdots, x_m) = \mathbb{E}_E \left[ \max_{w \in W} \left\| \frac{1}{m} \sum_{i=1}^m e_i \nabla_w l(w, x_i) \right\|_2 \right], \tag{30}$$

is the *empirical Rademacher complexity* and $E = \{e_i\}_i$ is a Rademacher sequence, namely, a sequence of independent random variables taking $\pm 1$ with equal probabilities. We can now revisit (26) and write that

$$\|w_t - w^\natural\|_2^2 \leq 4\alpha^2 (1 - \eta_\rho)^t \Delta_0 + \frac{2\alpha^2 \overline{\eta}_\rho}{\rho} + \frac{2\alpha^2 \Upsilon_{m,W}^2(\varepsilon)}{\zeta_L^2}, \tag{31}$$

which holds except with a probability of at most $e^{-\varepsilon}$. In addition, for every $w, w' \in W$, we may write that

$$
\begin{aligned}
\|\nabla L_m(w) &- \nabla L_m(w')\|_2 \\
&\leq \|\nabla L(w) - \nabla L(w')\|_2 + \|\nabla L_m(w) - \nabla L(w)\|_2 \\
&\quad + \|\nabla L_m(w') - \nabla L(w')\|_2 \quad \text{(triangle inequality)} \\
&\leq \sigma_L \|w - w'\|_2 + 2\Upsilon_{m,W}(\varepsilon), \quad \text{(see (20,29))} &\tag{32}
\end{aligned}
$$

except with a probability of at most $e^{-\varepsilon}$. Likewise, for every $w, w' \in W$, we have that

$$
\begin{aligned}
&\|\nabla L_m(w) - \nabla L_m(w')\|_2 \\
&\geq \|\nabla L_m(w) - \nabla L_m(w)\|_2 - \|\nabla L_m(w) - \nabla L(w)\|_2 \\
&\quad - \|\nabla L_m(w') - \nabla L(w')\|_2 \quad \text{(triangle inequality)} \\
&\geq \zeta_L \|w - w'\|_2 - 2\Upsilon_{m,W}(\varepsilon), \quad \text{(see (20,29))}
\end{aligned}
\tag{33}
$$

except with a probability of at most $e^{-\varepsilon}$. Therefore, $L_m$ satisfies the restricted strong convexity and smoothness in Definition 1 with

$$
\begin{aligned}
\mu_L &= \sigma_L, \qquad \nu_L = \zeta_L, \\
\overline{\mu}_L &= \overline{\zeta}_L = 2\Upsilon_{m,W}(\varepsilon).
\end{aligned}
\tag{34}
$$

Our findings in this section are summarized below.

**Theorem 4. (generalization error)** *Suppose that Assumptions 2-5 hold and recall that the training samples $\{x_i\}_{i=1}^m$ are drawn independently from the probability space $(\mathbb{X}, \chi)$ for a compact set $\mathbb{X} \subset \mathbb{R}^d$ with diameter $\mathrm{diam}(\mathbb{X})$.*

*For a set $W \subset \mathbb{R}^p$, let $R = 1_W$ be the indicator function on $W$, and set $H \equiv 0$ in (1). Suppose that solution $w^*$ of (1) belongs to the relative interior of $W$. For $\varepsilon > 0$, evaluate the quantities in Theorem 2 with*

$$
\begin{aligned}
\mu_L &= \sigma_L, \qquad \nu_L = \zeta_L, \\
\overline{\mu}_L &= \overline{\zeta}_L = 4\mathcal{R}_W(x_1, \cdots, x_m) \\
&\quad + 6\sigma_l \,\mathrm{diam}(\mathbb{X}) \sqrt{\frac{\varepsilon + 2}{2m}},
\end{aligned}
\tag{35}
$$

*where $\mathcal{R}_W(x_1, \cdots, x_m)$ in the empirical Rademacher complexity defined in (30). If the requirements on the step sizes in (22) hold, we then have that*

$$
\begin{aligned}
\|w_t - w^\natural\|_2^2 &\leq 4\alpha^2 (1 - \eta_\rho)^t \Delta_0 + \frac{2\alpha^2 \overline{\eta}_\rho}{\rho} + \frac{8\alpha^2}{\zeta_L^2} \mathcal{R}_W(x_1, \cdots, x_m)^2 \\
&\quad + \frac{18\alpha^2 \sigma_l^2 \,\mathrm{diam}(\mathbb{X})^2 (\varepsilon + 2)}{m},
\end{aligned}
\tag{36}
$$

*except with a probability of at most $e^{-\varepsilon}$.*

Most of the remarks about Theorem 1 also apply to Theorem 4 and we note that $\|w_t - w^\natural\|_2$ reduces by increasing the number of training samples $m$, before asymptotically reaching the generalization error

$$
2\psi_\rho + \frac{8}{\zeta_L^2} \mathcal{R}_W(x_1, \cdots, x_m)^2.
\tag{37}
$$

Computing the Rademacher complexity above for specific choices of the network structure and loss is itself potentially a complicated task, which we will not pursue by the virtue of the generality of our results so far. The key technical challenge there is computing the corresponding *entropy integral*, which involves estimating the *covering numbers* of the set $W$ Mohri et al. [2018]. One last takeaway point from the statistical accuracy in (37) is the following. If

$$
\overline{\eta}_\rho = O(\rho \cdot \mathcal{R}_W(x_1, \cdots, x_m)^2 / \zeta_L^2),
\tag{38}
$$

the asymptotic optimization error in Theorem 1 does not play an important role in determining the generalization error above. In words, if (38) holds, then Algorithm 1 converges to the ball of statistical accuracy around $w^\natural$. Here, $O$ stands for the standard Big-O notation.

# B Proof of Proposition 1

The feedforward network $G = G_\Xi : \mathbb{R}^s \to \mathbb{R}^d$ is a composition of linear maps and entry-wise applications of the activation functions, and hence is also of class $C^1$. Its Jacobian $DG : \mathbb{R}^s \to \mathbb{R}^{d \times s}$

is thus a continuous function and its restriction to the compact subset $\mathcal{D} \subseteq \mathbb{R}^s$ is Lipschitz-continuous. Therefore, there exists $\nu_G \geq 0$ such that

$$\|DG(z') - DG(z)\|_2 \leq \nu_G \|z' - z\|, \qquad \forall z, z' \in \mathcal{D}.$$

From standard arguments it then follows that Assumption 2 holds in the sense that

$$
\begin{aligned}
\|G(z') - G(z) - DG(z)(z' - z)\|_2 &= \left\| \int_0^1 (DG(tz' + (1-t)z) - DG(z))(z' - z) dt \right\|_2 \\
&\leq \int_0^1 \|DG(tz' + (1-t)z) - DG(z)\|_2 \|z' - z\|_2 dt \\
&\leq \nu_G \int_0^1 t \|z' - z\|^2 dt = \frac{\nu_G}{2} \|z' - z\|_2^2,
\end{aligned}
$$

for every $z, z' \in \mathbb{R}^s$.

In order to show that Assumption 3 (near-isometry) also holds, we will require the following simple fact:

**Lemma 5.** *Let $G : \mathcal{D} \subseteq \mathbb{R}^s \to \mathbb{R}^d$ have a left inverse $H : G(\mathcal{D}) \subseteq \mathbb{R}^d \to \mathbb{R}^s$ which is Lipschitz-continuous with constant $\iota_G > 0$. Then it holds that*

$$\frac{1}{\iota_G} \|z' - z\| \leq \|G(z') - G(z)\|, \qquad \forall z', z \in D.$$

*Proof.*

$$\|z' - z\| = \|H(G(z')) - H(G(z))\| \leq \iota_G \|G(z') - G(z)\|.$$

$\square$

We now proceed to show that Assumption 3 holds. We suppose $G$ is of the form

$$G(z) = \omega_k W_k (\omega_{k-1} W_{k-1} \ldots (\omega_1 W_1 z) \ldots),$$

for weight matrices $\{W_k\}_k$. First note that, by the compactness of the domain of $G$, the values of the hidden layers are always contained in a product of compact intervals, and so we can replace $\omega_i$ by its restriction to such sets. Each $\omega_i$ is continuous, defined on a product of intervals, and is stricly increasing so that they have a continuous left inverse $\omega_i^{-1}$ [Garling, 2014, Proposition 6.4.5]. The assumption of non-decreasing layer sizes implies that the $W_i$ are tall matrices of dimensions $(m_i, n_i)$ with $m_i \geq n_i$, whose columns are almost surely linearly independent after an arbitrarily small perturbation. In such case they have a left matrix inverse $W_i^{-1}$, which as a bounded linear map, is continuous. It then follows that $G$ has a continuous left inverse of the form

$$G^{-1} = W_1^{-1} \circ \omega_1^{-1} \ldots W_k^{-1} \circ \omega_k^{-1},$$

which is a continuous mapping and is defined on $G(\mathcal{D})$ which by continuity of $G$ is compact, hence $G^{-1}$ is Lipschitz-continuous. The result then follows by the Lipschitz continuity of the map $G$ (restricted to the compact domain $\mathcal{D}$) and Lemma 5.

## C  Proof of Theorem 1

It is convenient throughout the supplementary material to use a slightly different notation for Lagrangian, compared to the body of the paper. To improve the readability of the proof, let us list here the assumptions on the empirical loss $L$ and prior $G$ that are used throughout this proof. For every iteration $t$, we assume that

$$
\begin{aligned}
& L(w_t) - L(w^*) - \langle w_t - w^*, \nabla L(w^*) \rangle \\
& \geq \frac{\mu_L}{2} \|w_t - w^*\|_2^2, \qquad \text{(strong convexity of } L) 
\end{aligned}
\tag{39}
$$

$$
\begin{aligned}
& L(w_{t+1}) - L(w_t) - \langle w_{t+1} - w_t, \nabla L(w_t) \rangle \\
& \leq \frac{\nu_L}{2} \|w_{t+1} - w_t\|_2^2, \qquad \text{(strong smoothness of } L)
\end{aligned}
\tag{40}
$$

$$\|G(z') - G(z) - DG(z) \cdot (z' - z)\|_2$$
$$\leq \frac{\nu_G}{2}\|z' - z\|_2^2, \qquad \text{(strong smoothness of } G) \tag{41}$$

$$\iota_G\|z' - z\|_2 \leq \|G(z') - G(z)\|_2 \leq \kappa_G\|z' - z\|_2, \qquad \text{(near-isometry of } G) \tag{42}$$

$$\|DG(z) \cdot (z' - z)\|_2 \leq \kappa_G\|z' - z\|_2, \qquad \text{(Lipschitz continuty of } G) \tag{43}$$

For the sake of brevity, let us set

$$v = (w, z) \in \mathbb{R}^{p+s},$$

$$\mathcal{L}_\rho(v, \lambda) := \mathcal{L}_\rho(w, z, \lambda) := L(w) + R(w) + H(z) + \langle w - G(z), \lambda \rangle$$
$$+ \frac{\rho}{2}\|w - G(z)\|_2^2, \qquad \text{(augmented Lagrangian)} \tag{44}$$

$$\mathcal{L}'_\rho(v, \lambda) := \mathcal{L}'_\rho(w, z, \lambda) = L(w) + \langle w - G(z), \lambda \rangle + \frac{\rho}{2}\|w - G(z)\|_2^2, \tag{45}$$

$$A(v) = A(w, z) := w - G(z). \qquad \text{(feasibility gap)} \tag{46}$$

Let also $v^* = (w^*, z^*)$ be a solution of (1) and let $\lambda^*$ be a corresponding optimal dual variable. The first-order necessary optimality conditions for (1) are

$$\begin{cases} -\nabla_v \mathcal{L}'_\rho(v^*, \lambda^*) \in \partial R(w^*) \times \partial H(z^*), \\ w^* = G(z^*), \end{cases} \tag{47}$$

where $\partial R(w^*)$ and $\partial H(z^*)$ are the subdifferentials of $R$ and $H$, respectively, at $w^*$ and $z^*$. Throughout the proof, we will also often use the notation

$$\Delta_t := \mathcal{L}_\rho(v_t, \lambda_t) - \mathcal{L}_\rho(v^*, \lambda^*), \tag{48}$$

$$\Delta'_t := \mathcal{L}'_\rho(v_t, \lambda_t) - \mathcal{L}'_\rho(v^*, \lambda^*), \tag{49}$$

$$\delta_t := \|w_t - w^*\|_2, \qquad \delta'_t := \|z_t - z^*\|_2, \tag{50}$$

$$A_t := A(v_t) = w_t - G(z_t). \tag{51}$$

In particular, with this new notation, the dual update can be rewritten as

$$\lambda_{t+1} = \lambda_t + \sigma_{t+1}A_{t+1}. \qquad \text{(see Algorithm 1)} \tag{52}$$

First, in Appendix D, we control the smoothness of $\mathcal{L}'_\rho$ over the trajectory of the algorithm.

**Lemma 6.** *For every iteration t, it holds that*

$$\mathcal{L}'_\rho(w_{t+1}, z_{t+1}, \lambda_t) - \mathcal{L}'_\rho(w_t, z_{t+1}, \lambda_t) - \langle w_{t+1} - w_t, \nabla_w \mathcal{L}'_\rho(w_t, z_{t+1}, \lambda_t) \rangle$$
$$\leq \frac{\nu_\rho}{2}\|w_{t+1} - w_t\|_2^2, \tag{53}$$

$$\mathcal{L}'_\rho(w_t, z_{t+1}\lambda_t) - \mathcal{L}'_\rho(w_t, z_t, \lambda_t) - \langle z_{t+1} - z_t, \nabla_z \mathcal{L}'_\rho(w_t, z_t, \lambda_t) \rangle$$
$$\leq \frac{\xi_\rho}{2}\|z_{t+1} - z_t\|_2^2, \tag{54}$$

$$\|\nabla_w \mathcal{L}'_\rho(w_t, z_{t+1}, \lambda_t) - \nabla_w \mathcal{L}'_\rho(w_t, z_t, \lambda_t)\|_2 \leq \tau_\rho\|z_{t+1} - z_t\|_2^2, \tag{55}$$

*where*

$$\nu_\rho := \nu_L + \rho. \tag{56}$$

$$\xi_\rho := \nu_G(\lambda_{\max} + \rho \max_i \|A_i\|_2) + 2\rho\kappa_G^2, \tag{57}$$

$$\tau_\rho := \rho\kappa_G. \tag{58}$$

Second, in the following result we ensure that $\mathcal{L}_\rho$ and $\mathcal{L}'_\rho$ are sufficiently regular along the trajectory of our algorithm, see Appendix E for the proof.

**Lemma 7.** *For every iteration $t$, it holds that*

$$\Delta_t \geq \frac{\mu_\rho \delta_t^2}{2} + \frac{\mu'_\rho \delta_t'^2}{2} - \overline{\mu}_\rho, \tag{59}$$

$$\Delta'_t + \langle v^* - v_t, \nabla_v \mathcal{L}'_\rho(v_t) \rangle \leq \frac{\omega_\rho \delta_t^2}{2} + \frac{\omega'_\rho \delta_t'^2}{2}, \tag{60}$$

*where*

$$\mu_\rho := \mu_L - 2\rho, \qquad \mu'_\rho := \frac{\rho \iota_G^2}{2} - \nu_G \|\lambda^*\|_2, \tag{61}$$

$$\overline{\mu}_\rho := \frac{3}{\rho} \left( \lambda_{\max}^2 + \|\lambda^*\|_2^2 \right), \tag{62}$$

$$\omega_\rho := 0, \qquad \omega'_\rho := \frac{\nu_G}{2} \left( \lambda_{\max} + \rho \right). \tag{63}$$

Having listed all the necessary technical lemmas above, we now proceed to prove Theorem 1. Using the smoothness of $\mathcal{L}'_\rho$, established in Lemma 6, we argue that

$$
\begin{aligned}
& \mathcal{L}'_\rho(v_{t+1}, \lambda_{t+1}) \\
&= L(w_{t+1}) + \langle A_{t+1}, \lambda_{t+1} \rangle + \frac{\rho}{2} \|A_{t+1}\|_2^2 \qquad \text{(see (45))} \\
&= L(w_{t+1}) + \langle A_{t+1}, \lambda_t \rangle + \left( \frac{\rho}{2} + \sigma_{t+1} \right) \|A_{t+1}\|_2^2 \qquad \text{(see (52))} \\
&= \mathcal{L}'_\rho(w_{t+1}, z_{t+1}, \lambda_t) + \sigma_{t+1} \|A_{t+1}\|_2^2 \qquad \text{(see (44))} \\
&\leq \mathcal{L}'_\rho(w_t, z_{t+1}, \lambda_t) + \langle w_{t+1} - w_t, \nabla_w \mathcal{L}'_\rho(w_t, z_{t+1}, \lambda_t) \rangle + \frac{\nu_\rho}{2} \|w_{t+1} - w_t\|_2^2 \\
&\quad + \overline{\nu}_\rho + \sigma_{t+1} \|A_{t+1}\|_2^2 \qquad \text{(see (53))} \\
&\leq \mathcal{L}'_\rho(w_t, z_{t+1}, \lambda_t) + \langle w_{t+1} - w_t, \nabla_w \mathcal{L}'_\rho(w_t, z_{t+1}, \lambda_t) \rangle + \frac{1}{2\alpha} \|w_{t+1} - w_t\|_2^2 \\
&\quad + \overline{\nu}_\rho + \sigma_{t+1} \|A_{t+1}\|_2^2, \tag{64}
\end{aligned}
$$

where the last line above holds if the step size $\alpha$ satisfies

$$\alpha \leq \frac{1}{\nu_\rho}. \tag{65}$$

According to Algorithm 1, we can equivalently write the $w$ updates as

$$w_{t+1} = \arg\min_w \langle w - w_t, \nabla_w \mathcal{L}'_\rho(w_t, z_{t+1}, \lambda_t) \rangle + \frac{1}{2\alpha} \|w - w_t\|_2^2 + R(w). \tag{66}$$

In particular, consider above the choice of $w = \theta w^* + (1-\theta) w_t$ for $\theta \in [0,1]$ to be set later. We can then bound the last line of (64) as

$$
\begin{aligned}
& \mathcal{L}'_\rho(v_{t+1}, \lambda_{t+1}) + R(w_{t+1}) \\
&= \mathcal{L}'_\rho(w_t, z_{t+1}, \lambda_t) + \min_w \langle w - w_t, \nabla_w \mathcal{L}'_\rho(w_t, z_{t+1}, \lambda_t) \rangle \\
&\quad + \frac{1}{2\alpha} \|w - w_t\|_2^2 + R(w) + \sigma_{t+1} \|A_{t+1}\|_2^2 \qquad \text{(see (64,66))} \\
&\leq \mathcal{L}'_\rho(w_t, z_{t+1}, \lambda_t) + \theta \langle w^* - w_t, \nabla_w \mathcal{L}'_\rho(w_t, z_{t+1}, \lambda_t) \rangle + \frac{\theta^2 \delta_t^2}{2\alpha} \\
&\quad + \theta R(w^*) + (1-\theta) R(w_t) + \sigma_{t+1} \|A_{t+1}\|_2^2 \qquad \text{(convexity of } R) \\
&= \mathcal{L}'_\rho(w_t, z_{t+1}, \lambda_t) + \theta \langle w^* - w_t, \nabla_w \mathcal{L}'_\rho(w_t, z_t, \lambda_t) \rangle + \frac{\theta^2 \delta_t^2}{2\alpha} \\
&\quad + \theta \langle w^* - w_t, \nabla_w \mathcal{L}'_\rho(w_t, z_{t+1}, \lambda_t) - \nabla_w \mathcal{L}'_\rho(w_t, z_t, \lambda_t) \rangle \\
&\quad + \theta R(w^*) + (1-\theta) R(w_t) + \sigma_{t+1} \|A_{t+1}\|_2^2. \tag{67}
\end{aligned}
$$

The last inner product above can be controlled as

$$\theta\langle w^* - w_t, \nabla_w \mathcal{L}'_\rho(w_t, z_{t+1}, \lambda_t) - \nabla_w \mathcal{L}'_\rho(w_t, z_t, \lambda_t)\rangle$$

$$\leq \frac{\theta^2 \delta_t^2}{2\alpha} + \frac{\alpha}{2}\|\nabla_w \mathcal{L}'_\rho(w_t, z_{t+1}, \lambda_t) - \nabla_w \mathcal{L}'_\rho(w_t, z_t, \lambda_t)\|_2^2 \qquad (2\langle a, b\rangle \leq \|a\|_2^2 + \|b\|_2^2 \text{ and } (50))$$

$$\leq \frac{\theta^2 \delta_t^2}{2\alpha} + \alpha\tau_\rho^2\|z_{t+1} - z_t\|_2^2, \qquad \text{(see (55))} \tag{68}$$

which, after substituting in (67), yields that

$$\mathcal{L}'_\rho(v_{t+1}, \lambda_{t+1}) + R(w_{t+1})$$

$$\leq \mathcal{L}'_\rho(w_t, z_{t+1}, \lambda_t) + \theta\langle w^* - w_t, \nabla_w \mathcal{L}'_\rho(w_t, z_t, \lambda_t)\rangle + \frac{\theta^2 \delta_t^2}{\alpha}$$

$$+ \alpha\tau_\rho^2\|z_{t+1} - z_t\|_2^2 + \theta R(w^*) + (1-\theta)R(w_t) + \sigma_{t+1}\|A_{t+1}\|_2^2. \tag{69}$$

Regarding the right-hand side above, the smoothness of $\mathcal{L}'_\rho$ in Lemma 6 allows us to write that

$$\mathcal{L}'_\rho(w_t, z_{t+1}, \lambda_t) + \alpha\tau_\rho^2\|z_{t+1} - z_t\|_2^2$$

$$\leq \mathcal{L}'_\rho(w_t, z_t, \lambda_t) + \langle z_{t+1} - z_t, \nabla_z \mathcal{L}'_\rho(w_t, z_t, \lambda_t)\rangle$$

$$+ \left(\frac{\xi_\rho}{2} + \alpha\tau_\rho^2\right)\|z_{t+1} - z_t\|_2^2. \qquad \text{(see (54))} \tag{70}$$

If we assume that the primal step sizes $\alpha, \beta$ satisfy

$$\frac{\xi_\rho}{2} + \alpha\tau_\rho^2 \leq \frac{1}{2\beta}, \tag{71}$$

we can simplify (70) as

$$\mathcal{L}'_\rho(w_t, z_{t+1}, \lambda_t) + \alpha\tau_\rho^2\|z_{t+1} - z_t\|_2^2$$

$$\leq \mathcal{L}'_\rho(w_t, z_t, \lambda_t) + \langle z_{t+1} - z_t, \nabla_z \mathcal{L}'_\rho(w_t, z_t, \lambda_t)\rangle + \frac{1}{2\beta}\|z_{t+1} - z_t\|_2^2. \qquad \text{(see (71))} \tag{72}$$

From Algorithm 1, recall the equivalent expression of the $z$ updates as

$$z_{t+1} = \arg\min_z \langle z - z_t, \nabla_z \mathcal{L}'_\rho(w_t, z_t, \lambda_t)\rangle + \frac{1}{2\beta}\|z - z_t\|_2^2 + H(z), \tag{73}$$

and consider the choice of $z = \theta z^* + (1-\theta)z_t$ above, with $\theta \in [0, 1]$ to be set later. Combining (72,73) leads us to

$$\mathcal{L}'_\rho(w_t, z_{t+1}, \lambda_t) + \alpha\tau_\rho^2\|z_{t+1} - z_t\|_2^2 + H(z_{t+1})$$

$$= \mathcal{L}'_\rho(w_t, z_t, \lambda_t) + \min_z\langle z - z_t, \nabla_z \mathcal{L}'_\rho(w_t, z_t, \lambda_t)\rangle + \frac{1}{2\beta}\|z - z_t\|_2^2 + H(z) \qquad \text{(see (72,73))}$$

$$\leq \mathcal{L}'_\rho(w_t, z_t, \lambda_t) + \theta\langle z^* - z_t, \nabla_z \mathcal{L}'_\rho(w_t, z_t, \lambda_t)\rangle + \frac{\theta^2 \delta_t'^2}{2\beta} + H(\theta z^* + (1-\theta)z_t)$$

$$\leq \mathcal{L}'_\rho(w_t, z_t, \lambda_t) + \theta\langle z^* - z_t, \nabla_z \mathcal{L}'_\rho(w_t, z_t, \lambda_t)\rangle + \frac{\theta^2 \delta_t'^2}{2\beta}$$

$$+ \theta H(z^*) + (1-\theta)H(z_t). \qquad \text{(convexity of } H) \tag{74}$$

By combining (69,74), we reach

$$
\begin{aligned}
&\mathcal{L}_\rho(v_{t+1}, \lambda_{t+1}) \\
&= \mathcal{L}'_\rho(v_{t+1}, \lambda_{t+1}) + R(w_{t+1}) + H(z_{t+1}) \qquad \text{(see (44,45))} \\
&\leq \mathcal{L}'_\rho(w_t, z_{t+1}, \lambda_t) + \theta\langle w^* - w_t, \nabla_w \mathcal{L}'_\rho(w_t, z_t, \lambda_t)\rangle + \frac{\theta^2 \delta_t^2}{\alpha} + \alpha \tau_\rho^2 \|z_{t+1} - z_t\|_2^2 \\
&\quad + \theta R(w^*) + (1-\theta) R(w_t) + H(z_{t+1}) + \sigma_{t+1}\|A_{t+1}\|_2^2 \qquad \text{(see (69))} \\
&\leq \mathcal{L}'_\rho(v_t, \lambda_t) + \theta\langle v^* - v_t, \nabla_z \mathcal{L}'_\rho(v_t, \lambda_t)\rangle + \frac{\theta^2 \delta_t^2}{\alpha} + \frac{\theta^2 \delta_t^{'2}}{2\beta} \\
&\quad + \theta R(z^*) + (1-\theta) R(z_t) + \theta H(z^*) + (1-\theta) H(z_t) \\
&\quad + \sigma_{t+1}\|A_{t+1}\|_2^2 \qquad \text{(see (74))} \\
&= \mathcal{L}_\rho(v_t, \lambda_t) + \theta\langle v^* - v_t, \nabla_z \mathcal{L}'_\rho(v_t, \lambda_t)\rangle + \frac{\theta^2 \delta_t^2}{\alpha} + \frac{\theta^2 \delta_t^{'2}}{2\beta} \\
&\quad + \theta(R(z^*) + H(z^*) - R(z_t) - H(z_t)) + \sigma_{t+1}\|A_{t+1}\|_2^2 \qquad \text{(see (44,45))} \\
&\leq \mathcal{L}_\rho(v_t, \lambda_t) + \theta\left(\frac{\omega_\rho \delta_t^2}{2} + \frac{\omega'_\rho \delta_t^{'2}}{2} - \Delta'_t\right) + \frac{\theta^2 \delta_t^2}{\alpha} + \frac{\theta^2 \delta_t^{'2}}{2\beta} \\
&\quad + \theta(R(z^*) + H(z^*) - R(z_t) - H(z_t)) + \sigma_{t+1}\|A_{t+1}\|_2^2 \qquad \text{(see (60))} \\
&= \mathcal{L}_\rho(v_t, \lambda_t) + \theta\left(\frac{\omega_\rho \delta_t^2}{2} + \frac{\omega'_\rho \delta_t^{'2}}{2} - \Delta_t\right) + \frac{\theta^2 \delta_t^2}{\alpha} + \frac{\theta^2 \delta_t^{'2}}{2\beta} \\
&\quad + \sigma_{t+1}\|A_{t+1}\|_2^2 \qquad \text{(see (44,45))}
\end{aligned}
\tag{75}
$$

After recalling (48) and by subtracting $\mathcal{L}_\rho(v^*, \lambda^*)$ from both sides, (75) immediately implies that

$$
\begin{aligned}
\Delta_{t+1} &\leq \Delta_t + \frac{\omega_\rho \delta_t^2}{2} + \frac{\omega'_\rho \delta_t^{'2}}{2} + \theta\left(\overline{\omega}_\rho - \Delta_t\right) + \frac{\theta^2 \delta_t^2}{\alpha} + \frac{\theta^2 \delta_t^{'2}}{2\beta} \\
&\quad + \sigma_{t+1}\|A_{t+1}\|_2^2, \quad \text{(see (48,75))}
\end{aligned}
\tag{76}
$$

where we also used the assumption that $\theta \leq 1$ above. To remove the feasibility gap $\|A_{t+1}\|_2$ from the right-hand side above, we write that

$$
\begin{aligned}
\|A_{t+1}\|_2 &= \|w_{t+1} - G(z_{t+1})\|_2 \qquad \text{(see (51))} \\
&= \|w_{t+1} - w^* - (G(z_{t+1}) - G(z^*))\|_2 \qquad ((w^*, z^*) \text{ is a solution of (1))} \\
&\leq \|w_{t+1} - w^*\|_2 + \|G(z_{t+1}) - G(z^*)\|_2 \qquad \text{(triangle inequality)} \\
&\leq \|w_{t+1} - w^*\|_2 + \kappa_G \|z_{t+1} - z^*\|_2 \qquad \text{(see (42))} \\
&= \delta_{t+1} + \kappa_G \delta'_{t+1}, \qquad \text{(see (50))}
\end{aligned}
\tag{77}
$$

which, after substituting in (76), yields that

$$
\begin{aligned}
\Delta_{t+1} &\leq \Delta_t + \frac{\omega_\rho \delta_t^2}{2} + \frac{\omega'_\rho \delta_t^{'2}}{2} + \theta\left(\overline{\omega}_\rho - \Delta_t\right) + \frac{\theta^2 \delta_t^2}{\alpha} + \frac{\theta^2 \delta_t^{'2}}{2\beta} + 2\sigma_{t+1}\delta_{t+1}^2 + 2\sigma_{t+1}\kappa_G^2 \delta_{t+1}^{'2} \\
&\qquad \text{(see (77) and } (a+b)^2 \leq 2a^2 + 2b^2) \\
&\leq \Delta_t + \frac{\omega_\rho \delta_t^2}{2} + \frac{\omega'_\rho \delta_t^{'2}}{2} + \theta\left(\overline{\omega}_\rho - \Delta_t\right) + \frac{\theta^2 \delta_t^2}{\alpha} + \frac{\theta^2 \delta_t^{'2}}{2\beta} + 2\sigma_0 \delta_{t+1}^2 + 2\sigma_0 \kappa_G^2 \delta_{t+1}^{'2}. \\
&\qquad (\sigma_{t+1} \leq \sigma_0 \text{ in Algorithm 1})
\end{aligned}
\tag{78}
$$

For every iteration $t$, suppose that

$$
\frac{\delta_t^2}{\alpha} + \frac{\delta_t^{'2}}{\beta} \geq \overline{\eta}_\rho \geq \frac{\overline{\mu}_\rho}{\min\left(\frac{\alpha \mu_\rho}{4}, \frac{\beta \mu'_\rho}{2}\right) - \sqrt{\max\left(\frac{\alpha}{2}(\omega_\rho + 4\sigma_0), \beta(\omega'_\rho + 4\sigma_0 \kappa_G^2)\right)}},
\tag{79}
$$

for $\overline{\eta}_\rho$ to be set later. Consequently, it holds that

$$
\frac{\Delta_t}{\frac{2\delta_t^2}{\alpha} + \frac{\delta_t'^2}{\beta}} \geq \frac{\frac{\mu_\rho \delta_t^2}{2} + \frac{\mu_\rho' \delta_t'^2}{2} - \overline{\mu}_\rho}{\frac{2\delta_t^2}{\alpha} + \frac{\delta_t'^2}{\beta}} \qquad \text{(see (59))}
$$

$$
\geq \min\left(\frac{\alpha\mu_\rho}{4}, \frac{\beta\mu_\rho'}{2}\right) - \frac{\overline{\mu}_\rho}{\frac{2\delta_t^2}{\alpha} + \frac{\delta_t'^2}{\beta}}
$$

$$
\geq \min\left(\frac{\alpha\mu_\rho}{4}, \frac{\beta\mu_\rho'}{2}\right) - \frac{\overline{\mu}_\rho}{\overline{\eta}_\rho} \qquad \text{(see (79))}
$$

$$
\geq \sqrt{\max\left(\frac{\alpha}{2}\left(\omega_\rho + 4\sigma_0\right), \beta(\omega_\rho' + 4\sigma_0\kappa_G^2)\right)}. \qquad \text{(see (79))} \tag{80}
$$

We now set

$$
\widehat{\theta}_t := \min\left(\sqrt{\frac{\Delta_t^2}{\left(\frac{2\delta_t^2}{\alpha} + \frac{\delta_t'^2}{\beta}\right)^2} - \max\left(\frac{\alpha}{2}\left(\omega_\rho + 4\sigma_0\right), \beta\left(\omega_\rho' + 4\sigma_0\kappa_G^2\right)\right)}, 1\right), \tag{81}
$$

which is well-defined, as verified in (80). From (80,81), it also immediately follows that

$$
\widehat{\theta}_t \in [0, 1], \qquad \forall t, \tag{82}
$$

$$
\Delta_t \geq 0, \qquad \forall t, \tag{83}
$$

which we will use later on in the proof. Consider first the case where $\widehat{\theta}_t < 1$. To study the choice of $\theta = \widehat{\theta}_t$ in (76), we will need the bound

$$
- \widehat{\theta}_t \Delta_t + \widehat{\theta}_t^2 \left(\frac{\delta_t^2}{\alpha} + \frac{\delta_t'^2}{2\beta}\right)
$$

$$
= - \sqrt{\frac{\Delta_t^4}{\left(\frac{2\delta_t^2}{\alpha} + \frac{\delta_t'^2}{\beta}\right)^2} - \Delta_t^2 \max\left(\frac{\alpha}{2}\left(\omega_\rho + 4\sigma_0\right), \beta\left(\omega_\rho' + 4\sigma_0\kappa_G^2\right)\right)}
$$

$$
+ \frac{\Delta_t^2}{\frac{4\delta_t^2}{\alpha} + \frac{2\delta_t'^2}{\beta}} - \max\left(\frac{\alpha}{2}\left(\omega_\rho + 4\sigma_0\right), \beta\left(\omega_\rho' + 4\sigma_0\kappa_G^2\right)\right)\left(\frac{\delta_t^2}{\alpha} + \frac{\delta_t'^2}{2\beta}\right) \qquad \text{(see (83))}
$$

$$
\leq - \frac{\Delta_t^2}{\frac{4\delta_t^2}{\alpha} + \frac{2\delta_t'^2}{\beta}} + \Delta_t \sqrt{\max\left(\frac{\alpha}{2}\left(\omega_\rho + 4\sigma_0\right), \beta\left(\omega_\rho' + 4\sigma_0\kappa_G^2\right)\right)}
$$

$$
- \max\left(\frac{\alpha}{2}\left(\omega_\rho + 4\sigma_0\right), \beta\left(\omega_\rho' + 4\sigma_0\kappa_G^2\right)\right)\left(\frac{\delta_t^2}{\alpha} + \frac{\delta_t'^2}{2\beta}\right), \tag{84}
$$

where the inequality above uses $\sqrt{a - b} \geq \sqrt{a} - \sqrt{b}$. Substituting (84) back into (78), we reach

$$
\Delta_{t+1} \leq \Delta_t - \frac{\Delta_t^2}{\frac{4\delta_t^2}{\alpha} + \frac{2\delta_t'^2}{\beta}} + \Delta_t \sqrt{\max\left(\frac{\alpha}{2}\left(\omega_\rho + 4\sigma_0\right), \beta\left(\omega_\rho' + 4\sigma_0\kappa_G^2\right)\right)} \qquad \text{(see (78,84))}
$$

$$
\leq \Delta_t - \left(\min\left(\frac{\alpha\mu_\rho}{4}, \frac{\beta\mu_\rho'}{2}\right) - \frac{\overline{\mu}_\rho}{\overline{\eta}_\rho}\right)\frac{\Delta_t}{2}
$$

$$
+ \Delta_t \sqrt{\max\left(\frac{\alpha}{2}\left(\omega_\rho + 4\sigma_0\right), \beta\left(\omega_\rho' + 4\sigma_0\kappa_G^2\right)\right)} \qquad \text{(see third line of (80) and (83))}
$$

$$
\leq \left(1 - \min\left(\frac{\alpha\mu_\rho}{8}, \frac{\beta\mu_\rho'}{4}\right) + \frac{\overline{\mu}_\rho}{2\overline{\eta}_\rho} + \sqrt{\max\left(\frac{\alpha}{2}\left(\omega_\rho + 4\sigma_0\right), \beta\left(\omega_\rho' + 4\sigma_0\kappa_G^2\right)\right)}\right)\Delta_t
$$

$$
=: \eta_{\rho,1}\Delta_t, \qquad \text{if } \Delta_t < \frac{\delta_t^2}{\alpha} + \frac{\delta_t'^2}{\beta}. \tag{85}
$$

Next consider the case where $\widehat{\theta}_t = 1$. With the choice of $\theta = \widehat{\theta}_t = 1$ in (78), we find that

$$
\begin{aligned}
\Delta_{t+1} &\leq \left( \frac{\omega_\rho}{2} + \frac{1}{\alpha} + \rho \right) \delta_t^2 + \left( \frac{\omega_\rho'}{2} + \frac{1}{2\beta} + \rho \kappa_G^2 \right) \delta_t'^2 \qquad \text{(see (78))} \\
&\leq \frac{1}{2} \left( 1 + \max \left( \frac{\alpha}{2}(\omega_\rho + 4\sigma_0), \beta(\omega_\rho' + 4\sigma_0\kappa_G^2) \right) \right) \cdot \left( \frac{2\delta_t^2}{\alpha} + \frac{\delta_t'^2}{\beta} \right) \\
&\leq \frac{1}{2} \sqrt{1 + \max \left( \frac{\alpha}{2}(\omega_\rho + 4\sigma_0), \beta(\omega_\rho' + 4\sigma_0\kappa_G^2) \right)} \Delta_t \qquad \text{(see (81))} \\
&=: \eta_{\rho,2} \Delta_t, \qquad \text{if } \Delta_t \geq \frac{\delta_t^2}{\alpha} + \frac{\delta_t'^2}{\beta}. 
\end{aligned}
\tag{86}
$$

To simplify the above expressions, let us assume that

$$
\sqrt{\max \left( \frac{\alpha}{2}(\omega_\rho + 4\sigma_0), \beta(\omega_\rho' + 4\sigma_0\kappa_G^2) \right)} \leq \min \left( \frac{\alpha\mu_\rho}{16}, \frac{\beta\mu_\rho'}{8} \right) \leq \frac{1}{2},
\tag{87}
$$

from which it follows that

$$
\begin{aligned}
\max(\eta_{\rho,1}, \eta_{\rho,2}) &\leq 1 - \min \left( \frac{\alpha\mu_\rho}{16}, \frac{\beta\mu_\rho'}{8} \right) + \frac{\overline{\mu}_\rho}{2\overline{\eta}_\rho} \\
&\leq 1 - \min \left( \frac{\alpha\mu_\rho}{32}, \frac{\beta\mu_\rho'}{16} \right) \\
&=: 1 - \eta_\rho \in [0, 1),
\end{aligned}
\tag{88}
$$

where the second line above holds if

$$
\overline{\eta}_\rho \geq \frac{\overline{\mu}_\rho}{\min \left( \frac{\alpha\mu_\rho}{16}, \frac{\beta\mu_\rho'}{8} \right)}.
\tag{89}
$$

Then, by unfolding (85,86), we reach

$$
\Delta_t \leq (1 - \eta_\rho)^t \Delta_0.
\tag{90}
$$

Moreover, by combining (59,90), we can bound the error, namely,

$$
\begin{aligned}
\frac{\delta_t^2}{\alpha} + \frac{\delta_t'^2}{\beta} &\leq \max(\alpha\mu_\rho, \beta\mu_\rho') \left( \mu_\rho \delta_t^2 + \mu_\rho' \delta_t'^2 \right) \\
&\leq \mu_\rho \delta_t^2 + \mu_\rho' \delta_t'^2 \qquad \text{(see (65,71), Lemmas 6 and 7)} \\
&\leq 2(\Delta_t + \overline{\mu}_\rho) \qquad \text{(see (59))} \\
&\leq 2(1 - \eta_\rho)^t \Delta_0 + \frac{2\overline{\mu}_\rho}{\eta_\rho} \\
&\leq 2(1 - \eta_\rho)^t \Delta_0 + \frac{2\overline{\mu}_\rho}{\min \left( \frac{\alpha\mu_\rho}{16}, \frac{\beta\mu_\rho'}{8} \right)} \qquad \text{(see (88))} \\
&=: 2(1 - \eta_\rho)^t \Delta_0 + \frac{\overline{\eta}_\rho}{\rho}. \qquad \text{(this choice of } \overline{\eta}_\rho \text{ satisfies (79,89)).}
\end{aligned}
\tag{91}
$$

It remains to bound the feasibility gap $\|A_t\|_2$, see (51). Instead of (77), we consider the following alternative approach to bound $\|A_t\|_2$. Using definition of $\Delta_t$ in (48), we write that

$$
\begin{aligned}
\Delta_t &= \mathcal{L}_\rho(v_t, \lambda_t) - \mathcal{L}_\rho(v^*, \lambda^*) \qquad \text{(see (48))} \\
&= \mathcal{L}_\rho(v_t, \lambda_t) - \mathcal{L}_\rho(v_t, \lambda^*) + \mathcal{L}_\rho(v_t, \lambda^*) - \mathcal{L}_\rho(v^*, \lambda^*) \\
&= \langle A_t, \lambda_t - \lambda^* \rangle + \mathcal{L}(v_t, \lambda^*) - \mathcal{L}(v^*, \lambda^*) + \frac{\rho}{2} \|A_t\|_2^2,
\end{aligned}
\tag{92}
$$

where

$$
\mathcal{L}(v, \lambda) = \mathcal{L}(w, z, \lambda) := L(w) + R(w) + H(z) + \langle w - G(z), \lambda \rangle.
\tag{93}
$$

It is not difficult to verify that $\mathcal{L}(v^*, \lambda^*) = \mathcal{L}_\rho(v^*, \lambda^*)$ is the optimal value of problem (1) and that $\mathcal{L}(v_t, \lambda^*) \geq \mathcal{L}(v^*, \lambda^*)$, from which it follows that

$$\Delta_t \geq \langle A_t, \lambda_t - \lambda^* \rangle + \frac{\rho}{2} \|A_t\|_2^2 \qquad \text{(see (92))}$$

$$\geq -\frac{\rho}{4} \|A_t\|_2^2 - \frac{1}{\rho} \|\lambda_t - \lambda^*\|_2^2 + \frac{\rho}{2} \|A_t\|_2^2 \qquad \text{(Holder's inequality and } 2ab \leq a^2 + b^2)$$

$$\geq -\frac{2}{\rho} \|\lambda_t\|_2^2 - \frac{2}{\rho} \|\lambda^*\|_2^2 + \frac{\rho}{4} \|A_t\|_2^2 \qquad ((a+b)^2 \leq 2a^2 + 2b^2)$$

$$\geq -\frac{2\lambda_{\max}^2}{\rho} - \frac{2\|\lambda^*\|_2^2}{\rho} + \frac{\rho}{4} \|A_t\|_2^2, \qquad \text{(see (100))} \tag{94}$$

which, in turn, implies that

$$\|A_t\|_2^2 \leq \frac{4}{\rho} \left( \Delta_t + \frac{2\lambda_{\max}^2}{\rho} + \frac{2\|\lambda^*\|_2^2}{\rho} \right) \qquad \text{(see (94))}$$

$$\leq \frac{4}{\rho} \left( (1 - \eta_\rho)^t \Delta_0 + \frac{2(\overline{\eta}_{\rho,1} + \overline{\eta}_{\rho,2})}{\eta_\rho} + \frac{2\lambda_{\max}^2}{\rho} + \frac{2\|\lambda^*\|_2^2}{\rho} \right) \qquad \text{(see (90))}$$

$$\leq \frac{4}{\rho} \left( (1 - \eta_\rho)^t \Delta_0 + \frac{\overline{\eta}_\rho + 2\lambda_{\max}^2 + 2\|\lambda^*\|_2^2}{\rho} \right) \qquad \text{(see (91))}$$

$$=: \frac{4(1 - \eta_\rho)^t \Delta_0}{\rho} + \frac{\widetilde{\eta}_\rho}{\rho^2}. \tag{95}$$

This completes the proof of Theorem 1.

Let us also inspect the special case where $\mu_L \gg \rho \gtrsim 1$ and $\iota_G^2 \gg \nu_G$, where $\approx$ and $\gtrsim$ suppress any universal constants and dependence on the dual optimal variable $\lambda^*$, for the sake of simplicity. From Lemmas 6 and 7, it is easy to verify that

$$\nu_\rho \approx \nu_L, \qquad \xi_\rho \approx \rho \kappa_G^2, \qquad \tau_\rho = \rho \kappa_G,$$

$$\mu_\rho \approx \mu_L, \qquad \mu_\rho' \approx \rho \iota_G^2, \qquad \overline{\mu}_\rho \approx \rho^{-1}, \qquad \omega_\rho' \approx \rho \nu_G. \tag{96}$$

We can then take

$$\alpha \approx \frac{1}{\nu_L}, \qquad \text{(see (65))}$$

$$\beta \approx \frac{1}{\xi_\rho} \approx \frac{1}{\rho \kappa_G^2}, \qquad \text{(see (71))}$$

$$\eta_\rho \approx \min\left( \frac{\mu_L}{\nu_L}, \frac{\iota_G^2}{\kappa_G^2} \right), \qquad \text{(see (88))}$$

$$\overline{\eta}_\rho \approx \frac{\rho \overline{\mu}_\rho}{\min\left( \alpha \mu_\rho, \beta \mu_\rho' \right)} \approx \max\left( \frac{\nu_L}{\mu_L}, \frac{\kappa_G^2}{\iota_G^2} \right), \qquad \text{(see (91))}$$

$$\widetilde{\eta}_\rho \approx \overline{\eta}_\rho \approx \max\left( \frac{\nu_L}{\mu_L}, \frac{\kappa_G^2}{\iota_G^2} \right). \qquad \text{(see (95))} \tag{97}$$

Lastly, for (87) to hold, it suffices that

$$\sigma_0 \lesssim \rho \min\left( \frac{\mu_L^2}{\nu_L^2}, \frac{\iota_G^4}{\kappa_G^4} \right) =: \sigma_{0,\rho}. \tag{98}$$

# D Proof of Lemma 6

To prove (53), we write that

$$
\begin{aligned}
&\mathcal{L}'_\rho(w_{t+1}, z_{t+1}, \lambda_t) - \mathcal{L}'_\rho(w_t, z_{t+1}, \lambda_t) - \langle w_{t+1} - w_t, \nabla_w \mathcal{L}'_\rho(w_t, z_{t+1}, \lambda_t) \rangle \\
&= L(w_{t+1}) - L(w_t) - \langle w_{t+1} - w_t, \nabla_w L(w_t) \rangle \\
&\quad + \frac{\rho}{2}\|w_{t+1} - G(z_{t+1})\|_2^2 - \frac{\rho}{2}\|w_t - G(z_{t+1})\|_2^2 - 2\rho\langle w_{t+1} - w_t, w_t - G(z_{t+1}) \rangle \qquad \text{(see (45))}\\
&\leq \frac{\nu_L}{2}\|w_{t+1} - w_t\|_2^2 + \bar{\nu}_L + \frac{\rho}{2}\|w_{t+1} - w_t\|_2^2 \qquad \text{(see (40))}\\
&=: \frac{\nu_\rho}{2}\|w_{t+1} - w_t\|_2^2 + \bar{\nu}_\rho. 
\end{aligned} \tag{99}
$$

To prove (54), let us first control the dual sequence $\{\lambda_t\}_t$ by writing that

$$
\begin{aligned}
\|\lambda_t\|_2 &= \left\|\lambda_0 + \sum_{i=1}^t \sigma_i A_i\right\|_2 \qquad \text{(see (52))}\\
&\leq \|\lambda_0\|_2 + \sum_{i=1}^t \sigma_i \|A_i\|_2 \qquad \text{(triangle inequality)}\\
&\leq \|\lambda_0\|_2 + \sum_{t'=1}^t \frac{\sigma_0}{i \log^2(i+1)} \\
&\leq \|\lambda_0\|_2 + c\sigma_0 \\
&=: \lambda_{\max},
\end{aligned} \tag{100}
$$

where

$$
c \geq \sum_{t=1}^\infty \frac{1}{t \log^2(t+1)}. \tag{101}
$$

We now write that

$$
\begin{aligned}
&\mathcal{L}'_\rho(w_t, z_{t+1}, \lambda_t) - \mathcal{L}'_\rho(w_t, z_t, \lambda_t) - \langle z_{t+1} - z_t, \nabla_z \mathcal{L}'_\rho(w_t, z_t, \lambda_t) \rangle \\
&= -\langle G(z_{t+1}) - G(z_t) - DG(z_t)(z_{t+1} - z_t), \lambda_t \rangle \\
&\quad + \frac{\rho}{2}\|w_t - G(z_{t+1})\|_2^2 - \frac{\rho}{2}\|w_t - G(z_t)\|_2^2 \\
&\quad + \rho\langle DG(z_t)(z_{t+1} - z_t), w_t - G(z_t) \rangle. \qquad \text{(see (45))}
\end{aligned} \tag{102}
$$

To bound the first inner product on the right-hand side above, we write that

$$
\begin{aligned}
&\langle G(z_{t+1}) - G(z_t) - DG(z_t)(z_{t+1} - z_t), \lambda_t \rangle \\
&\leq \|G(z_{t+1}) - G(z_t) - DG(z_t)(z_{t+1} - z_t)\|_2 \cdot \|\lambda_t\|_2 \qquad \text{(Cauchy-Shwartz's inequality)}\\
&\leq \frac{\nu_G \lambda_{\max}}{2}\|z_{t+1} - z_t\|_2^2 \qquad \text{(see (41,100))}
\end{aligned} \tag{103}
$$

The remaining component on the right-hand side of (102) can be bounded as

$$
\begin{aligned}
&\|w_t - G(z_{t+1})\|_2^2 - \|w_t - G(z_t)\|_2^2 + 2\langle DG(z_t)(z_{t+1} - z_t), w_t - G(z_t) \rangle \\
&= \|w_t - G(z_{t+1})\|_2^2 - \|w_t - G(z_t)\|_2^2 + 2\langle G(z_{t+1}) - G(z_t), w_t - G(z_t) \rangle \\
&\quad - 2\langle G(z_{t+1}) - G(z_t) - DG(z_t)(z_{t+1} - z_t), w_t - G(z_t) \rangle \\
&= \|G(z_{t+1}) - G(z_t)\|_2^2 \\
&\quad + 2\langle G(z_{t+1}) - G(z_t) - DG(z_t)(z_{t+1} - z_t), w_t - G(z_t) \rangle \\
&\leq \|G(z_{t+1}) - G(z_t)\|_2^2 \\
&\quad + 2\|G(z_{t+1}) - G(z_t) - DG(z_t)(z_{t+1} - z_t)\|_2 \cdot \|w_t - G(z_t)\|_2 \qquad \text{(Cauchy-Shwartz's inequality)}\\
&\leq \kappa_G^2\|z_{t+1} - z_t\|_2^2 + \nu_G\|z_{t+1} - z_t\|_2^2\|w_t - G(z_t)\|_2 \qquad \text{(see (41,42))}\\
&\leq \kappa_G^2\|z_{t+1} - z_t\|_2^2 + \nu_G\|z_{t+1} - z_t\|_2^2 \max_i \|A_i\|_2. \qquad \text{(see (51))}
\end{aligned} \tag{104}
$$

Substituting the bounds in (103,104) back into (102), we find that

$$
\begin{aligned}
&\mathcal{L}'_\rho(w_t, z_{t+1}, \lambda_t) - \mathcal{L}'_\rho(w_t, z_t, \lambda_t) - \langle z_{t+1} - z_t, \nabla_z \mathcal{L}'_\rho(w_t, z_t, \lambda_t) \rangle \\
&\leq \frac{1}{2} \left( \nu_G(\lambda_{\max} + \rho \max_i \|A_i\|_2) + \rho \kappa_G^2 \right) \|z_{t+1} - z_t\|_2^2 \\
&=: \frac{\xi_\rho}{2} \|z_{t+1} - z_t\|_2^2 + \overline{\xi}_\rho,
\end{aligned} \tag{105}
$$

which proves (54). To prove (55), we write that

$$
\begin{aligned}
&\|\nabla_w \mathcal{L}'_\rho(w_t, z_{t+1}, \lambda_t) - \nabla_w \mathcal{L}'_\rho(w_t, z_t, \lambda_t)\|_2 \\
&= \rho \|G(z_{t+1}) - G(z_t)\|_2 \qquad \text{(see (45))} \\
&\leq \rho \kappa_G \|z_{t+1} - z_t\|_2 \qquad \text{(see (42))} \\
&=: \tau_\rho \|z_{t+1} - z_t\|_2 + \overline{\tau}_\rho.
\end{aligned} \tag{106}
$$

This completes the proof of Lemma 6.

# E  Proof of Lemma 7

For future reference, we record that

$$
\begin{aligned}
&\langle v_t - v^*, \nabla_v \mathcal{L}'_\rho(v^*) \rangle \\
&= \langle w_t - w^*, \nabla_w \mathcal{L}'_\rho(v^*) \rangle + \langle z_t - z^*, \nabla_z \mathcal{L}'_\rho(v^*) \rangle \qquad (v = (w, z)) \\
&= \langle w_t - w^*, \nabla L(w^*) + \lambda^* + \rho(w^* - G(z^*)) \rangle - \langle DG(z^*)(z_t - z^*), \lambda^* + \rho(w^* - G(z^*)) \rangle \qquad \text{(see (45))} \\
&= \langle w_t - w^*, \nabla L(w^*) + \lambda^* \rangle - \langle DG(z^*)(z_t - z^*), \lambda^* \rangle, \qquad \text{(see (47))} \tag{107}
\end{aligned}
$$

where the last line above uses the feasibility of $v^*$ in (1). To prove (59), we use the definition of $\mathcal{L}_\rho$ in (44) to write that

$$
\begin{aligned}
&\mathcal{L}_\rho(v_t, \lambda_t) - \mathcal{L}_\rho(v^*, \lambda^*) \\
&= \mathcal{L}'_\rho(v_t, \lambda_t) - \mathcal{L}'_\rho(v^*, \lambda^*) + R(w_t) - R(w^*) + L(z_t) - L(z^*) \qquad \text{(see (44,45))} \\
&\geq \mathcal{L}'_\rho(v_t, \lambda_t) - \mathcal{L}'_\rho(v^*, \lambda^*) - \langle v_t - v^*, \nabla_v \mathcal{L}'_\rho(v^*, \lambda^*) \rangle \qquad \text{(see (47))} \\
&= L(w_t) - L(w^*) - \langle w_t - w^*, \nabla L(u^*) \rangle \\
&\quad + \langle A_t, \lambda_t \rangle - \langle w_t - w^* - DG(z^*)(z_t - z^*), \lambda^* \rangle + \frac{\rho}{2} \|A_t\|_2^2 \qquad \text{(see (107))} \\
&\geq \frac{\mu_L \delta_t^2}{2} + \langle A_t, \lambda_t - \lambda^* \rangle + \frac{\rho}{2} \|A_t\|_2^2 \\
&\quad + \langle G(z_t) - G(z^*) - DG(z^*)(z_t - z_k^*), \lambda^* \rangle \qquad \text{(see (39,50))} \\
&\geq \frac{\mu_L \delta_t^2}{2} + \langle A_t, \lambda_t - \lambda^* \rangle + \frac{\rho}{2} \|A_t\|_2^2 - \frac{\nu_G \delta_t'^2}{2} \|\lambda^*\|_2. \qquad \text{(see (41,50))} \tag{108}
\end{aligned}
$$

To control the terms involving $A_t$ in the last line above, we write that

$$
\begin{aligned}
&\langle A_t, \lambda_t - \lambda^* \rangle + \frac{\rho}{2} \|A_t\|_2^2 \\
&= \frac{\rho}{2} \left\| A_t - \frac{\lambda_t - \lambda^*}{\rho} \right\|_2^2 - \frac{\|\lambda_t - \lambda^*\|_2^2}{2\rho} \\
&= \frac{\rho}{2} \left\| w_t - w^* - (G(z_t) - G(z^*)) - \frac{\lambda_t - \lambda^*}{\rho} \right\|_2^2 - \frac{\|\lambda_t - \lambda^*\|_2^2}{2\rho} \qquad \text{(see (47,51))} \\
&\geq \frac{\rho}{4} \|G(z_t) - G(z^*)\|_2^2 - \rho \delta_t^2 - \frac{3\|\lambda_t - \lambda^*\|_2^2}{2\rho} \qquad \left( \|a - b - c\|_2^2 \geq \frac{\|a\|_2^2}{2} - 2\|b\|_2^2 - 2\|c\|_2^2 \right) \\
&\geq \frac{\rho \iota_G^2 \delta_t'^2}{4} - \rho \delta_t^2 - \frac{3\|\lambda_t - \lambda^*\|_2^2}{2\rho} \qquad \text{(see (50,42))} \\
&\geq \frac{\rho \iota_G^2 \delta_t'^2}{4} - \rho \delta_t^2 - \frac{3}{\rho}(\lambda_{\max}^2 + \|\lambda^*\|_2^2), \qquad ((a+b)^2 \leq 2a^2 + 2b^2 \text{ and (100)}) \tag{109}
\end{aligned}
$$

which, after substituting in (108), yields that

$$
\begin{aligned}
&\mathcal{L}_\rho(v_t, \lambda_t) - \mathcal{L}_\rho(v^*, \lambda^*) \\
&\geq \frac{\mu_L - 2\rho}{2}\delta_t^2 + \frac{1}{2}\left(\frac{\rho\iota_G^2}{2} - \nu_G\|\lambda^*\|_2\right)\delta_t'^2 - \frac{3}{\rho}\left(\lambda_{\max}^2 + \|\lambda^*\|_2^2\right) \\
&\geq \frac{\mu_\rho\delta_t^2}{2} + \frac{\mu_\rho'\delta_t'^2}{2} - \overline{\mu}_\rho,
\end{aligned}
\tag{110}
$$

where

$$
\mu_\rho := \mu_L - 2\rho, \qquad \mu_\rho' := \frac{\rho\iota_G^2}{2} - \nu_G\|\lambda^*\|_2,
\tag{111}
$$

$$
\overline{\mu}_\rho := \frac{3}{\rho}\left(\lambda_{\max}^2 + \|\lambda^*\|_2^2\right).
\tag{112}
$$

This proves (59). To prove (60), we use the definition of $\mathcal{L}_\rho'$ in (45) to write that

$$
\begin{aligned}
&\mathcal{L}_\rho'(v^*, \lambda^*) - \mathcal{L}_\rho'(v_t, \lambda_t) - \langle v^* - v_t, \nabla_v \mathcal{L}_\rho'(v_t, \lambda_t)\rangle \\
&= L(w^*) - L(w_t) - \langle w^* - w_t, \nabla L(w_t)\rangle \\
&\quad - \langle A_t + DA(v_t)(v^* - v_t), \lambda_t\rangle \\
&\quad - \frac{\rho}{2}\langle A_t + 2DA(v_t)(v^* - v_t), A_t\rangle, \qquad \text{(see (45))}
\end{aligned}
\tag{113}
$$

where

$$
DA(v) = \begin{bmatrix} I_d & -DG(z) \end{bmatrix},
\tag{114}
$$

is the Jacobian of the map $A$. The second inner product on the right-hand side of (113) can be bounded as

$$
\begin{aligned}
&-\langle A_t + DA(v_t)(v^* - v_t), \lambda_t\rangle \\
&= -\langle w_t - G(z_t) + (w^* - w_t) - DG(z_t)(z^* - z_t), \lambda_t\rangle \qquad \text{(see (51,114))} \\
&= -\langle G(z^*) - G(z_t) - DG(z_t)(z^* - z_t), \lambda_t\rangle \qquad (w^* = G(z^*)) \\
&\geq -\frac{\nu_G\delta_t'^2}{2}\|\lambda_t\|_2 \qquad \text{(see (41,50))} \\
&\geq -\frac{\nu_G\delta_t'^2}{2}\lambda_{\max}. \qquad \text{(see (100))}
\end{aligned}
\tag{115}
$$

To control the last inner product on the right-hand side of (113), we write that

$$
\begin{aligned}
&-\frac{\rho}{2}\langle A_t + 2DA(v_t)(v^* - v_t), A_t\rangle \\
&= \frac{\rho}{2}\|A_t\|_2^2 - \rho\langle A_t + DA(v_t)(v^* - v_t), A_t\rangle \\
&\geq -\rho\|A_t + DA(v_t)(v^* - v_t)\|_2\|A_t\|_2 \qquad \text{(Holder's inequality)} \\
&= -\rho\|(w^* - G(z^*)) - (w_t - G(z_t)) - (w^* - w_t) + DG(z_t)(z^* - z_t)\|_2 \qquad \text{(see (51,114) and } w^* = G(z^*)) \\
&= -\rho\|G(z^*) - G(z_t) - DG(z_t)(z^* - z_t)\|_2 \\
&\geq -\frac{\rho\nu_G}{2}\|z^* - z_t\|_2^2 \qquad \text{(see (41))} \\
&= -\frac{\rho\nu_G\delta_t'^2}{2}. \qquad \text{(see (50))}
\end{aligned}
\tag{116}
$$

By substituting the bounds in (115,116) back into (113) and also using the convexity of $L$, we reach

$$
\begin{aligned}
&\mathcal{L}_\rho'(v^*, \lambda^*) - \mathcal{L}_\rho'(v_t, \lambda_t) - \langle v^* - v_t, \nabla_v \mathcal{L}_\rho'(v_t, \lambda_t)\rangle \\
&\geq -\frac{\nu_G}{2}\left(\lambda_{\max} + \rho\right)\delta_t'^2.
\end{aligned}
\tag{117}
$$

This proves (60), thus completing the proof of Lemma 7.

## F    Relation with Gradient Descent

Throughout this section, we set $R \equiv 0$ and $H \equiv 0$ in problem (1) and consider the updates in Algorithm 2, namely,

$$
\begin{aligned}
z_{t+1} &= z_t - \beta \boldsymbol{\nabla}_z \mathcal{L}_\rho(w_t, z_t, \lambda_t), \\
w_{t+1} &\in \operatorname*{argmin}_w \mathcal{L}_\rho(w, z_{t+1}, \lambda_t), \\
\lambda_{t+1} &= \lambda_t + \sigma_{t+1}(w_{t+1} - G(z_{t+1})).
\end{aligned}
\tag{118}
$$

From (2), recall that $\mathcal{L}_\rho(w, z, \lambda)$ is convex in $w$ and the second step in (118) is therefore often easy to implement with any over-the-shelf standard convex solver. Recalling (2), note also that the optimality condition for $w_{t+1}$ in (118) is

$$
w_{t+1} - G(z_t) = -\frac{1}{\rho}(\nabla L_m(w_{t+1}) + \lambda_t).
\tag{119}
$$

Using (2) again, we also write that

$$
\begin{aligned}
\nabla_z \mathcal{L}_\rho(w_{t+1}, z_t, \lambda_t) \\
&= -DG(z_t)^\top (\lambda_t + \rho(w_{t+1} - G(z_t))) \\
&= -DG(z_t)^\top (\lambda_t - \lambda_{t-1} - \nabla L_m(w_t)) \\
&= -DG(z_t)^\top (\sigma_t(w_t - G(z_t)) - \nabla L(w_t)),
\end{aligned}
\tag{120}
$$

where the last two lines above follow from (119,118), respectively. Substituting back into the $z$ update in (118), we reach

$$
z_{t+1} = z_t + \beta \sigma_t DG(z_t)^\top (w_t - G(z_t)) - \beta \nabla L(w_t) \qquad \text{(see (118,120))},
\tag{121}
$$

from which it follows that

$$
\begin{aligned}
\|z_{t+1} - (z_t - \beta \nabla L(G(z_t)))\|_2 \\
&\leq \beta \sigma_t \|DG(z_t)^\top (w_t - G(z_t))\|_2 + \beta \|\nabla L(w_t) - \nabla L(G(z_t))\|_2 \qquad \text{(see (121))} \\
&\leq \beta (\sigma_t \kappa_G + \nu_L) \|w_t - G(z_t)\|_2. \qquad \text{(see Assumptions 1 and 3)}
\end{aligned}
\tag{122}
$$

That is, as the feasibility gap vanishes in (24) in Theorem 1, the updates of Algorithm 2 match those of GD.

## G    Proof of Lemma 3

Recall that $R = 1_W$ and $H \equiv 0$ for this proof. Using the optimality of $w^* \in \operatorname{relint}(W)$ in (17), we can write that

$$
\begin{aligned}
\|\nabla L(w^*)\|_2 &\leq \|\nabla L_m(w^*)\|_2 + \|\nabla L_m(w^*) - \nabla L(w^*)\|_2 \qquad \text{(triangle inequality)} \\
&= \|\nabla L_m(w^*) - \nabla L(w^*)\|_2 \qquad (\nabla L_m(w^*) = 0) \\
&\leq \max_{w \in W} \|\nabla L_m(w) - \nabla L(w)\|_2.
\end{aligned}
\tag{123}
$$

On the other hand, using the strong convexity of $L$ in (20), we can write that

$$
\begin{aligned}
\|w^\natural - w^*\|_2 &\leq \frac{1}{\zeta_L} \|\nabla L(w^\natural) - \nabla L(w^*)\|_2 \qquad \text{(see (20))} \\
&= \frac{1}{\zeta_L} \|\nabla L(w^*)\| \qquad (\nabla L(w^\natural) = 0) \\
&\leq \frac{1}{\zeta_L} \max_{w \in W} \|\nabla L_m(w) - \nabla L(w)\|_2, \qquad \text{(see (123))}
\end{aligned}
\tag{124}
$$

which completes the proof of Lemma 3.

## H Experimental Setup Details

### H.1 Per-Iteration Computational Complexity

The gradient of the function

$$h(z) = \frac{1}{2}\|AG(z) - b\|_2^2 \tag{125}$$

follows the formula

$$\nabla h(z) = \nabla G(z) A^\top (AG(z) - b) \tag{126}$$

which involves one forward pass through the network $G$, in order to compute $G(z)$, as well as one backward pass to compute $\nabla G(z)$, and finally matrix-vector products to compute the final result.

On the other hand our ADMM first computes the iterate $z_{t+1}$ with gradient descent on the augmented lagrangian (2) as

$$z_{t+1} = z_t - \beta \nabla_z \mathcal{L}_\rho(w_t, z_t, \lambda_t) = -\nabla G(z_t)\lambda_t^\top - \rho \nabla G(z_t)(w_t - G(z_t))^\top \tag{127}$$

which involves one forward and one backward pass on the network $G$, as well as matrix-vector products. Then we perform the exact minimization procedure on the $w$ variable, which requires recomputing $G(z)$ on the new iterate $z_{t+1}$, involving one forward pass through the network, as well as the matrix-vector operations as described before. Recomputing the quantity $w_{t+1} - G(z_{t+1})$ is immediate upon which the dual stepsize $\sigma_{t+1}$ can be computed at negligible cost. Finally the dual variable update reads as

$$\lambda_{t+1} = \lambda_t + \sigma(w_{t+1} - G(z_{t+1})) \tag{128}$$

which involves only scalar products and vector additions of values already computed. All in all each GD iteration involves one forward and one backward pass, while ADMM computes two forward and one backward pass. Both algorithms require a few additional matrix-vector operations of similar complexity. For networks with multiple large layers, as usually encountered in practice, the complexity per iteration can then be estimated as the number of forward and backward passes, which are of similar complexity.

### H.2 Parameter Tuning

We run a grid search for the gradient descent (GD) algorithm In order to do so we fix a number of iterations and compare the average objective function over a batch of 100 random images and choose the best performing parameters. We repeat the tuning in all possible escenarios in the experiments. The results figures 4 - 5 (GD, Compressive sensing setup).

Figure 4: Performance of GD on the compressive sensing task for different step sizes. MNIST dataset. 156 (top) and 313 (bottom) linear measurements.

### H.3 Fast Exact Augmented Lagrangian Minimization with Respect to Primal Variable $w$

In the compressive sensing setup, the augmented lagrangian takes the form

$$\mathcal{L}_\rho(w, z, \lambda) := \frac{1}{2}\|Aw - b\|_2^2 + \langle \lambda, w - G(z) \rangle + \frac{\rho}{2}\|w - G(z)\|_2^2 \tag{129}$$

Figure 5: Performance of GD on the compressive sensing task for different step sizes. CelebA dataset. 2457 (top) and 4915 (bottom) linear measurements.

with respect to $w$, this is a strongly convex function which admits a unique minimizer given by the first order optimality condition

$$\nabla_w \mathcal{L}_\rho(w, z, \lambda) = A^\top(Aw - b) + \lambda + \rho(w - G(z)) = 0 \tag{130}$$

with solution

$$w^* = (A^\top A + \rho I)^{-1}(-\lambda + G(z) + A^\top b) \tag{131}$$

Given the SVD of $A = USV^\top$ we have $A^\top A = VDV^\top$, where $D$ corresponds to the diagonal matrix with the eigenvalues of $A^T A$. We then have that $A^\top A + \rho I = V(D + \rho I)V^\top$ so that

$$w^* = V(D + \rho I)V^\top(-\lambda + G(z) + A^\top b) \tag{132}$$

which involves only a fixed number of matrix-vector products per-iteration.

### H.4   Per-Iteration Computational Complexity

The gradient of the function

$$h(z) = \frac{1}{2}\|AG(z) - b\|_2^2 \tag{133}$$

follows the formula

$$\nabla h(z) = \nabla G(z)A^\top(AG(z) - b) \tag{134}$$

which involves one forward pass through the network $G$, in order to compute $G(z)$, as well as one backward pass to compute $\nabla G(z)$, and finally matrix-vector products to compute the final result.

On the other hand our ADMM first computes the iterate $z_{t+1}$ with gradient descent on the augmented lagrangian (129)

$$z_{t+1} = z_t - \beta \nabla_z \mathcal{L}_\rho(w_t, z_t, \lambda_t) = -\nabla G(z_t)\lambda_t^\top - \rho \nabla G(z_t)(w_t - G(z_t))^\top \tag{135}$$

which involves one forward and one backward pass on the network $G$, as well as matrix-vector products. Then we perform the exact minimization procedure on the $w$ variable, as described in H.3, which requires recomputing $G(z)$ on the new iterate $z_{t+1}$, involving one forward pass through the network, as well as the matrix-vector operations as described before. Recomputing the quantity $w_{t+1} - G(z_{t+1})$ is immediate upon which the dual stepsize $\sigma_{t+1}$ can be computed at negligible cost. Finally the dual variable update reads as

$$\lambda_{t+1} = \lambda_t + \sigma(w_{t+1} - G(z_{t+1})) \tag{136}$$

which involves only scalar products and vector additions of values already computed. All in all each GD iteration involves one forward and one backward pass, while ADMM computes two forward and one backward pass. Both algorithms require a few additional matrix-vector operations of similar complexity. For networks with multiple large layers, as usually encountered in practice, the complexity per iteration can then be estimated as the number of forward and backward passes, which are of similar complexity.

# I  Pseudocode for Algorithm 2

---

**Algorithm 2** Multi-scale Linearized ADMM

---

**Input:**  Differentiable $L$, proximal-friendly convex regularizers $R$ and $H$, differentiable prior $G$, penalty weight $\rho > 0$, primal step sizes $\alpha, \beta > 0$, initial dual step size $\sigma_0 > 0$, primal initialization $w_0$ and $z_0$, dual initialization $\lambda_0$, stopping threshold $\tau_c > 0$, iterations parameter $n$.

1  $z_{0,0} \leftarrow z_0, w_{0,0} \leftarrow w_0$

2  **for** k=0,..., K **do**

3  $\quad \rho_k \leftarrow \rho 2^k, \alpha_k \leftarrow \alpha 2^{-k}, \beta_k \leftarrow \beta 2^{-k}$

4  $\quad z_0 \leftarrow z_{0,k}, w_0 \leftarrow w_{0,k}$

5  $\quad$ **for** $t = 0, 1, \ldots, 2^k n$ **do**

6  $\quad\quad z_{t+1} \leftarrow \mathrm{P}_{\beta_k H}\left(z_t - \beta_k \nabla_z \mathcal{L}_{\rho_k}(w_t, z_t, \lambda_t)\right)$  $\hfill$ (primal updates)

7  $\quad\quad w_{t+1} \leftarrow \mathrm{P}_{\alpha_k R}\left(w_t - \alpha_k \nabla_w \mathcal{L}_{\rho}(w_t, z_{t+1}, \lambda_t)\right)$

8  $\quad\quad \sigma_{t+1} \leftarrow \min\left(\sigma_0, \dfrac{\sigma_0}{\|w_{t+1} - G(z_{t+1})\|_2 \, t \log^2(t+1)}\right)$  $\hfill$ (dual step size)

9  $\quad\quad \lambda_{t+1} \leftarrow \lambda_t + \sigma_{t+1}(w_{t+1} - G(z_{t+1}))$  $\hfill$ (dual update)

10  $\quad\quad s \leftarrow \dfrac{\|z_{t+1} - z_t\|_2^2}{\alpha_k} + \dfrac{\|w_{t+1} - w_t\|_2^2}{\beta_k} + \sigma_t \|w_t - G(z_t)\|_2^2 \leq \tau_c$  $\hfill$ (stopping criterion)

11  $\quad\quad$ **if** $s \leq \tau_c$ **then return** $(w_{t+1}, z_{t+1})$

12  $\quad\quad (w_{0,k+1}, z_{0,k+1}) \leftarrow (w_{t+1}, z_{t+1})$

13  $\quad$ **return** $(w_{0,K+1}, z_{0,K+1})$

---

## Footnotes

[1]If necessary, the inclusion $\{w_t\}_{t \ge 0} \subset W$ might be enforced by adding the indicator function of the convex hull of $W$ to $R$ in (1), similar to Agarwal et al. [2010].

[2]To be complete, $1_W(w) = 0$ if $w \in W$ and $1_W(w) = \infty$ otherwise.