[Reviews · NeurIPS 2019]

Reviewer 1



Post author feedback: I have read the comments from the author feedback and I maintain that this paper is well written and makes valuable contributions. I also think that the experimental analysis is sufficient. ====== The authors investigate ADMM in the context of solving optimization problems which consider generative priors. Convergence guarantees for the problem of inverse imaging under generative priors have been limited and restricted to gradient descent based schemes prior to this work [Bora '17, Hegde '18]. The presentation of the paper is excellent, and reads well; citations to prior work are adequate. The authors highlight three key assumptions required to show convergence of the ADMM algorithm to a O(1/rho) radius of the true solution: restricted strong convexity of loss function L, strong smoothness and near-isometry of the generator G. These assumptions are fairly standard in literature, and papers which have previously used them have been referred to appropriately. This paper also additionally establishes theoretical guarantees for gradient descent when used for the problem setting of the paper. Experiments in Sections 1 and 2 also highlight faster running times and epoch complexity, of ADMM with respect to conventional Gradient Descent, which is interesting. Minor: [126] typo "architectures" [777] !

Reviewer 2



I think the "global optimization" aspect of the main result and the fast (i.e., linear) convergence rate are very interesting, and perhaps also surprising. For example, for the least square problem min_z ||A G(z) - b|| prior works such as Hand & Voroninski [2017] and Heckel et al [2019] have established the global optimization aspect of simple gradient descent like algorithms. But the result obtained in this paper is much more general, and also applies to formulations with extra nonsmooth terms, with a practical numerical method. Moreover, general understanding of ADMM applied to nonconvex problems is still very rare. I think this result is definitely a beautiful addition to this line of literature also. Some minor comments: * While the generative prior idea is connected to the generator network in GAN, I think it's potentially confusing to emphasize GAN in the subsequent generative-prior induced optimization problems, i.e., saying things such as "This GAN-based optimization problem" (also, "with the explosion of GAN in popularity... and (possibly) non-smooth problems") because this could very likely lead people to think of the min-max problem involved in GAN. So probably after the brief recap of GAN, one can stick to terms like deep generative model to avoid such confusion. * It's interesting to see how the near-isometry property of G comes into play to yield the result. Suppose R = H = 0, can the main intuition of the proof be given, or maybe milestone steps that reflect how the property of G is used be sketched? I wonder how much the difference is to the convex case.

Reviewer 3



After rebuttal: Although I would expect more technical argument instead of providing formal justification, I am satisfied with the authors' feedback in general. I agree with the other two reviewers the theory part is excellent and the work of this paper is far more meaningful than minor issues. === Originality: The topic picked by the authors is distinct and the proposed algorithm is novel. Related work is adequately cited. Significance: The proposed algorithm is different from previous work. The comparison with gradient descent shows the proposed algorithm has a great advantage. However, I was expecting more results to demonstrate the efficiency of their algorithm. Clarity and quality: In general, this is a well-written paper with strong theory proof. My concern comes from the completeness of the paper. First, the paper would be more complete if a precise description of Algorithm 2 was provided like Algorithm 1. Second, I couldn't find any conclusion part to summarize the work in the paper, which should be given shortly. Third, the format of references does not meet NeurIPS style. This is a fancy and interesting paper. The theoretical analysis is very strong but it slightly lacks empirical results. The completeness of the paper could be improved. Although there are minor issues, the paper is worth an acceptance.

[Author Response · NeurIPS 2019]

We have addressed the minor issues pointed out by the reviewers; thank you for these constructive comments and for the encouraging remarks.

In particular, we have corrected some typos as pointed out by R1 and R3. As suggested by R2, we have also replaced the term GAN in some places with "generative models", to emphasize that our work is not directly related to the training of GANs. We wish to leave the reference style intact, as the guidelines allow such a choice. We will also clarify the x-axis units (time in seconds for figure 2, and cpu time of 1 iteration of gradient descent in figure 1).

In addition, it appears to us that R3's score does not match his/her positive review and we wish to ask R3 to consider revising his/her score. More precisely, R3 is mostly concerned with the experiments and empirical evidence presented in the work, although he/she agrees the paper is strong.

We believe that the theory is backed with solid numerical evidence, arguably well beyond what is common in similar papers, please see for example (1).

In particular, the CelebA dataset is much more complex than MNIST or CIFAR. The version we have used consists of RGB 64x64 images, which is 16 times the dimension of MNIST and 4 times the dimension of CIFAR. Moreover the generator network we used here is a Residual Network, considered to be a good baseline. We will consider also adding CIFAR10/100 examples to the final version, as well as a discussion on future research directions.

Please note that in our second experiment with MNIST, which is widely considered an easy dataset, both Gradient descent and Adam performed poorly. This was observed along a wide range of step sizes for each algorithm. This confidently suggests that these two algorithms would also perform poorly on more complex datasets. Our proposed linearized ADMM, on the other hand, consistently shows superior performance in such nonsmooth problem.

Finally, we also would like to highlight the following: As part of the requirements for this edition of NeurIPS, we will provide a polished code. While we can try to be as exhaustive as possible regarding the numerical evidence, we believe that the real impact will be much more than just including one additional dataset in the current paper.

For clarity, we will also add Algorithm 2 in explicit form to the appendix, as the space in the main text is quite limited. We will also add the proof sketch for Theorem 1 to further clarify the role of the near-isometry property of $G$; in short, when $G$ is near-isometric, the nonconvex problem locally resembles a convex problem.

We hope that we have addressed any concerns about this work and again wish to ask R3 to consider revising his/her score.

## References

[1] V. Shah and C. Hegde, "Solving Linear Inverse Problems Using GAN Priors: An Algorithm with Provable Guarantees," *arXiv:1802.08406 [cs, stat]*, Feb. 2018. arXiv: 1802.08406.


[Meta-Review · NeurIPS 2019]

This paper proposes a linearized ADMM method to solve inverse problems with generative priors, ie convex objectives subject to non-convex constraints. The constraint is parametrized by a generative model G that is assumed to be differentiable but otherwise highly non-convex. This setting is very relevant in current practice for various linear inverse problems. The proposed linearized ADMM has linear rate which is quite interesting. Overall this paper obtains interesting results for ADMM for nonconvex problems in a novel and relevant setting.